# BACH1 controls hepatic insulin signaling and glucose homeostasis in mice

Jiayu Jin[1,5], Yunquan He[1,5], Jieyu Guo [1,5], Qi Pan[1], Xiangxiang Wei[1], Chen Xu[2], Zhiyuan Qi[1], Qinhan Li[1], Siyu Ma[1], Jiayi Lin[1], Nan Jiang[1], Jinghua Ma[1], Xinhong Wang[1], Lindi Jiang[1], Qiurong Ding [3], Elena Osto[4], Xiuling Zhi[1] ✉ & Dan Meng [1] ✉

Hepatic insulin resistance is central to the metabolic syndrome. Here we investigate the role of BTB and CNC homology 1 (BACH1) in hepatic insulin signaling. BACH1 is elevated in the hepatocytes of individuals with obesity and patients with non-alcoholic fatty liver disease (NAFLD). Hepatocyte-specific *Bach1* deletion in male mice on a high-fat diet (HFD) ameliorates hyperglycemia and insulin resistance, improves glucose homeostasis, and protects against steatosis, whereas hepatic overexpression of *Bach1* in male mice leads to the opposite phenotype. BACH1 directly interacts with the protein-tyrosine phosphatase 1B (PTP1B) and the insulin receptor β (IR-β), and loss of BACH1 reduces the interaction between PTP1B and IR-β upon insulin stimulation and enhances insulin signaling in hepatocytes. Inhibition of PTP1B significantly attenuates BACH1-mediated suppression of insulin signaling in HFD-fed male mice. Hepatic BACH1 knockdown ameliorates hyperglycemia and improves insulin sensitivity in diabetic male mice. These results demonstrate a critical function for hepatic BACH1 in the regulation of insulin signaling and glucose homeostasis.

Insulin resistance is an early manifestation of numerous metabolic abnormalities and diseases, including obesity, type 2 diabetes mellitus, cardiovascular disease, and liver disease such as non-alcoholic steatohepatitis[1]. The liver is a major player in the maintenance of glucose and lipid homeostasis[2,3], through multiple insulin-mediated pathways[4]. Indeed, insulin increases glycogen synthesis by PI3K/AKT-dependent glycogen synthase kinase-3β (GSK-3β) deactivation and inhibits gluconeogenesis by hepatic phosphorylating transcription factor forkhead box protein O1 (FOXO1), thus maintaining blood glucose levels within a relatively narrow range[5]. On the contrary, insulin suppression of hepatic glucose output is impaired in insulin resistance associated with type 2 diabetes and obesity leading to enhanced

gluconeogenesis[1]. A better comprehension of the molecular mechanisms that drive hepatic insulin resistance may lead to crucial therapeutic approaches in metabolic diseases[6].

Protein tyrosine phosphatases (PTPs) play an essential role in insulin signaling by dephosphorylating the insulin receptor-β (IR-β) within minutes to inhibit IR-β-mediated insulin signaling[7,8]. Protein Tyrosine Phosphatase 1B (PTP1B), encoded by the protein tyrosine phosphatase non-receptor type 1 (PTPN1) gene, is an abundant enzyme expressed in all insulin-responsive tissues including livers[9]. PTP1B is localized on the cytoplasmic face of the endoplasmic reticulum (ER) and can access and dephosphorylate IR-β during endocytosis, biosynthesis, and by the ER network movement in close proximity to the

[1]Department of Physiology and Pathophysiology, School of Basic Medical Sciences, Department of Rheumatology, Zhongshan Hospital, Fudan University, Shanghai 200032, China. [2]Department of Pathology, Zhongshan Hospital, Fudan University, Shanghai 200032, China. [3]CAS Key Laboratory of Nutrition, Metabolism and Food Safety, Shanghai Institute of Nutrition and Health, University of Chinese Academy of Sciences, Chinese Academy of Sciences, Shanghai 200031, China. [4]Division of Physiology and Pathophysiology, Otto Loewi Research Center for Vascular Biology, Immunology and Inflammation, Medical University of Graz, Graz, Austria. [5]These authors contributed equally: Jiayu Jin, Yunquan He, Jieyu Guo. ✉e-mail: zhixiuling@fudan.edu.cn; dmeng@fudan.edu.cn

plasma membrane[10]. It was reported that CD36 deficiency impaired hepatic insulin signaling by enhancing the interaction of PTP1B with IR-β[11]. Previous studies have demonstrated that global PTP1B deficiency increased insulin sensitivity in the liver and muscle of mice and protected against weight gain and insulin resistance induced by a high-fat diet (HFD)[12,13]. Thus, PTP1B may represent a promising target for treating obesity-associated insulin resistance. However, so far, no PTP1B inhibitors entered III phase clinical trials due to the low bioavailability of many potent candidates and the competition effect with T-cell protein tyrosine phosphatase (TCPTP) which has much structural similarity to PTP1B[14]. Moreover, the specific regulatory mechanisms of PTP1B in the control of insulin signaling have yet to be determined.

Transcription factor BTB and CNC homology 1 (BACH1) is known to regulate multiple physiological and pathophysiological processes including oxidative stress[15], mitochondrial metabolism[16], cancer metastasis[17,18], ferroptosis[19], erythropoiesis[19], and immunity[20]. The N-terminal region of BACH1 contains a BTB domain, which functions as a protein interaction motif, while the C-terminal Bzip domain binds to DNA and mediates the heterodimerization of BACH1 with small Maf proteins[21]. BACH1 nuclear export was triggered by heme and cadmium, and both heme- and cadmium-induced BACH1 nuclear export signals were dependent on chromosome region maintenance 1 (CRM1)[22,23]. Our studies have demonstrated that deletion of endothelial BACH1 attenuated atherosclerosis by reducing endothelial inflammation[24], whereas BACH1 impaired the angiogenic response to peripheral ischemic injury in adult mice and induced endothelial cell (EC) apoptosis and cell cycle arrest[15,25]. BACH1 expression was upregulated and acted as a master regulator of aging-related genes in ECs of coronary arteries from aged monkeys, which concomitantly showed a downregulation of insulin-dependent signaling[26]. BACH1 orchestrated the metabolic and oxidative stress responses to palmitate and contributed to palmitate-induced human pancreatic β-cell dysfunction[27]. In turn, Bach1 deficiency in mice protected pancreatic β-cells from oxidative stress-induced apoptosis[28]. The global Bach1 knockout (KO) exerted hepatoprotective effects in methionine-choline deficient diet mice[29]. Instead, hepatocyte-specific sirtuin 6 deletion predisposed to non-alcoholic steatohepatitis via elevated BACH1 expression[30]. Therefore, although BACH1 seems to be associated with metabolic disorders, its specific role in hepatic insulin signaling and glucose homeostasis is unknown.

The present study explored the role of BACH1 in insulin resistance in the livers of HFD-fed or db/db diabetic mice. Using gain-of-function and loss-of-function approaches, we demonstrated that BACH1 impaired hepatic insulin signaling by enhancing the interaction of PTP1B with IR-β. Our findings revealed a crucial role for BACH1 in regulating insulin signaling, suggesting that BACH1 may become a promising target for the treatment of metabolic disorders such as obesity and type 2 diabetes mellitus.

## Results

### BACH1 is elevated in the livers of individuals with obesity, patients with NAFLD, and obese mice

To evaluate whether BACH1 plays a role in metabolic disorders, we first analyzed BACH1 mRNA expression from the published single-cell transcriptome sequencing data of the liver tissue of lean individuals or individuals with obesity (GEO accession no. GSE192742)[31]. Notably, the expression of BACH1 was higher in the hepatocytes of subjects with obesity than in lean subjects (Fig. 1a). We then evaluated liver samples from patients (men and women) with non-alcoholic fatty liver disease (NAFLD) by real-time quantitative PCR (qRT-PCR) (Fig. 1b), Western blot (Fig. 1c and Supplementary Fig. 1a) and immunohistochemistry (Fig. 1d and Fig. 1e) and found increased BACH1 expression in the liver tissues from patients with NAFLD compared with normal liver tissues, accompanied by the reduced glycogen content (periodic acid-Schiff

(PAS) staining) in the livers of patients with NAFLD (Fig. 1d). The sections of the livers of NAFLD patients incubated with rabbit IgG (10 μg/mL) were used as a negative control for BACH1 staining (Supplementary Fig. 1b). We next analyzed Bach1 mRNA expression from the published single-cell transcriptome sequencing data of the liver tissue of mice following a high-sucrose-and-high-fat diet (HSD) (GEO accession no. GSE182365)[32]. Indeed, Bach1 transcript was higher in the hepatocytes of mice fed with the HSD than in those fed with the chow diet (CD) (Fig. 1f). We then detected BACH1 mRNA and protein levels in the liver tissues of HFD-induced obesity mice, ob/ob mice, and db/db diabetic mice. Mice fed with HFD for 12 weeks were obese and hyperglycemic (Supplementary Fig. 3b and Fig. 2b). The mRNA (Fig. 1g) and protein levels (Fig. 1h and Supplementary Fig. 1c) of BACH1 were significantly increased in the livers of HFD-fed mice compared with the CD mice. Similarly, hepatic BACH1 protein levels were higher in ob/ob mice or in db/db diabetic mice than in WT mice (Fig. 1i, j and Supplementary Fig. 1d, e). Increased BACH1 expression was also observed in primary hepatocytes incubated with oleic acid (OA, 1 mM) (Fig. 1k and Supplementary Fig. 1f) or palmitic acid (PA, 1 mM) for 12 h (Supplementary Fig. 1g). These data suggested that hepatic expression of BACH1 is significantly upregulated in the livers of patients with NAFLD and obese mice.

### Hepatic BACH1 deficiency improves insulin signaling and dysregulation of glucose homeostasis in HFD-fed mice

We then determined the role of hepatic BACH1 on liver metabolism in mice. Eight-week-old male Bach1^flox/flox mice (20 g) were injected with AAV8-TBG-Cre, generating hepatocyte-deficient Bach1 (Bach1^LKO) mice (n = 8). Eight-week-old male Bach1^flox/flox mice (20 g) injected with AAV8-TBG-GFP were used as controls (n = 8) (Supplementary Fig. 2a). The deletion of the Bach1 gene in mouse liver was validated by Western blot and immunohistochemistry. The BACH1 protein was reduced by about 78% in whole-liver lysates from Bach1^LKO mice, while BACH1 protein expression in the heart and muscle was unperturbed (Supplementary Fig. 2b). The residual BACH1 expression in whole-liver lysates from Bach1^LKO mice was likely from non-parenchymal cells, including hepatic stellate cells, kupffer cells, sinusoidal endothelial cells, and pit cells. The immunohistochemical staining showed that about 76% of hepatocytes showed no BACH1 staining, especially in the portal area and surrounding areas of the livers from Bach1^LKO mice (Supplementary Fig. 2c). We then fed male Bach1^LKO mice and the controls with HFD or CD for 12 weeks, starting from the age of ten weeks (Supplementary Fig. 3a). Body weight, food intake, levels of fasting blood glucose and fasting insulin, glucose tolerance, and insulin tolerance were comparable between Bach1^LKO and the control mice maintained on the standard CD (Supplementary Fig. 3b–c and Fig. 2a–d). As expected, HFD feeding significantly increased body weights and levels of fasting blood glucose of both Bach1^LKO and the control mice compared with CD feeding (Supplementary Fig. 3b and Fig. 2b, left). Although the HFD Bach1^LKO mice showed no difference compared with the control mice in total body weight or food intake, the liver weight and liver weight/body weight ratios were significantly lower in HFD-fed Bach1^LKO compared to the control mice (Fig. 2a and Supplementary Fig. 3b–c). Levels of the fasting blood glucose, fasting insulin, and homeostasis model assessment of insulin resistance (HOMA-IR) index were significantly decreased in HFD-fed Bach1^LKO mice with respect to the control mice (Fig. 2b). Consistently, knockout of hepatic BACH1 in mice improved glucose tolerance and ameliorated insulin resistance compared with the control mice under HFD conditions, as indicated by the glucose tolerance test (GTT) (Fig. 2c) and the insulin tolerance test (ITT) (Fig. 2d). Impaired glycogen synthesis is a major contributor to hyperglycemia and insulin resistance[33]. A significantly higher glycogen content was observed in HFD-fed Bach1^LKO mice by PAS staining (Fig. 2e) and hepatocyte glycogen content analysis (Fig. 2f) compared to the HFD-fed control mice. Relevant to the

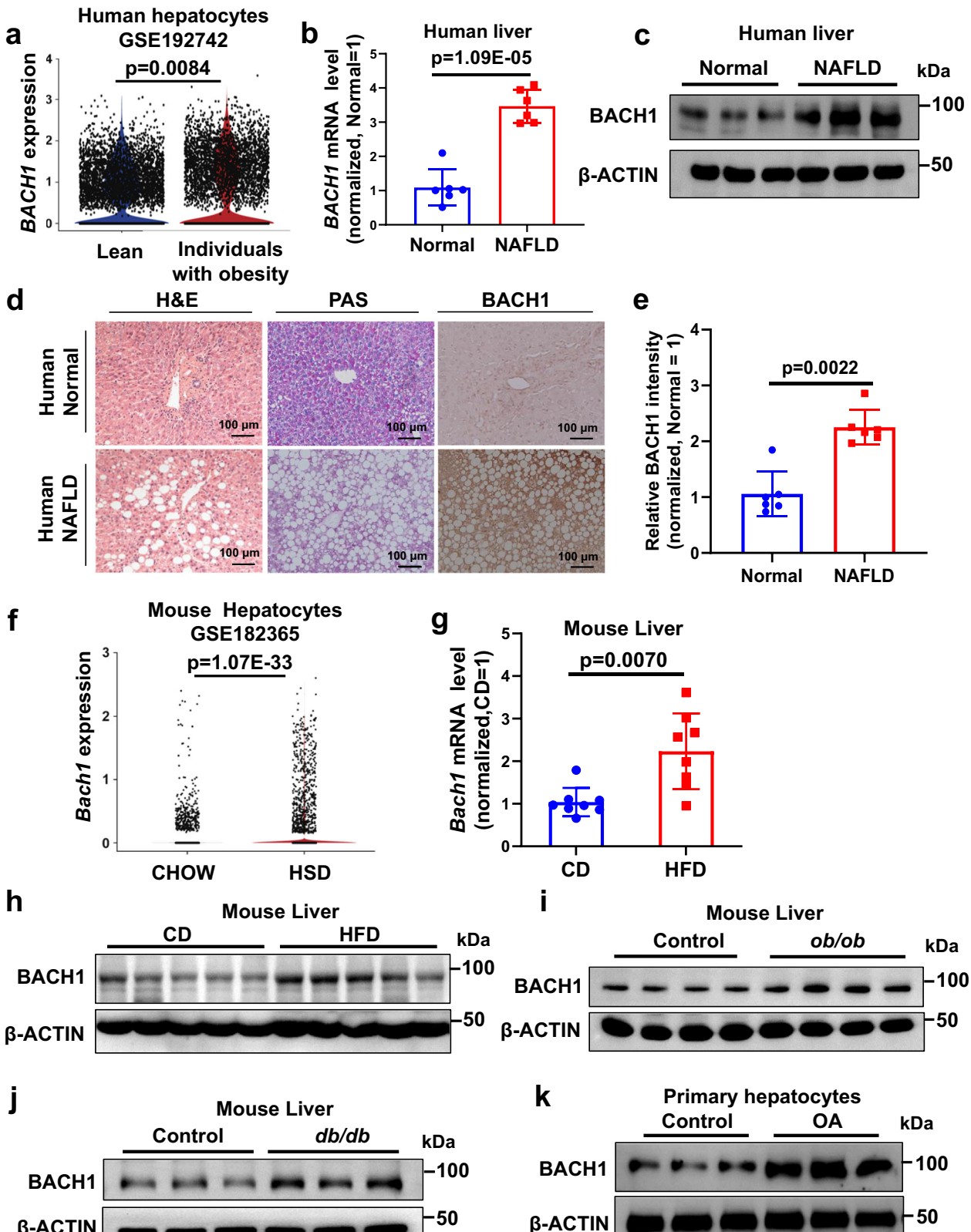

glycogen alterations, the mRNA levels of gluconeogenesis genes, phosphoenolpyruvate carboxykinase1 (*Pck1*), and a glucose-6-phosphatase catalytic subunit (*G6pc*) were reduced in the livers of *Bach1*[LKO] mice compared with the control mice on HFD (Fig. 2g). In addition, the levels of insulin-stimulated phosphorylated IR-β, AKT, GSK-3β, and FOXO1 were higher in the liver tissues of HFD-fed *Bach1*[LKO] mice than those of the control mice (Fig. 2h). Furthermore, HFD-fed

*Bach1*[LKO] mice showed significantly better hepatic function, as indicated by lower serum levels of alanine aminotransferase (ALT), aspartate aminotransferase (AST), and alkaline phosphatase (ALP) (Supplementary Fig. 3d–f). Hepatic *Bach1* deletion also resulted in a marked reduction in serum TC, TG, and NEFA levels compared with the control mice on HFD (Supplementary Fig. 3g–i). Consistently, HFD-fed *Bach1*[LKO] mice decreased the severity of hepatic steatosis compared

**Fig. 1 | BACH1 is elevated in the livers of individuals with obesity, patients with NAFLD, and obese mice. a** *BTB and CNC homology 1* (*BACH1*) mRNA expression in the hepatocytes of lean (*n* = 10) individuals and individuals with obesity (*n* = 4) from the published single-cell transcriptome sequencing database (GEO accession no. GSE192742). **b, c** *BACH1* mRNA expression (**b**) and protein expression (**c**) in the liver tissues from healthy subjects (Normal) and non-alcoholic fatty liver disease (NAFLD) patients (*n* = 6). **d** The representative images of hematoxylin and eosin (H&E), periodic acid-Schiff (PAS) staining, and immunohistochemistry assay of BACH1 from liver tissues of healthy subjects and NAFLD patients (scale bar = 100 μm, *n* = 6 individuals per group). **e** Quantitative data of immunohistochemistry assay of BACH1 in the liver tissues (*n* = 6). **f** *Bach1* mRNA expression in liver cells from mice following a high-sugar diet (HSD) compared with chow diet (CD) from the published single-cell transcriptome sequencing database (GEO accession no. GSE182365). **g, h** *Bach1* mRNA (*n* = 8) (**g**) and protein expression (*n* = 5) (**h**) in the liver tissues from male CD- and high-fat diet (HFD)-fed mice. **i, j** BACH1 protein expression in male *ob/ob* mice (*n* = 4) (**i**) and *db/db* mice (*n* = 3) (**j**). **k** Primary hepatocytes were treated with oleic acid (OA, 1 mM) for 12 h and then subjected to immunoblot analyses to determine the BACH1 protein expression (*n* = 3). Statistical analysis was performed by Wilcoxon Rank Sum test for (**a**) and (**f**), by two-tailed Mann–Whitney *U* test for (**e**), (**g**), and by unpaired two-tailed Student's *t*-test for (**b**). Data are presented as mean values ± SD. Source data are provided as a Source Data file.

with HFD-fed control mice, as evidenced by hematoxylin and eosin (H&E) and Oil Red O staining of liver sections (Supplementary Fig. 3j, k). Altogether, these data indicated that hepatocyte-specific deletion *Bach1* improved insulin signaling and dysregulation of glucose homeostasis in HFD-fed mice, and protected from HFD-induced liver steatosis.

## Hepatic overexpression of *Bach1* aggravates HFD-induced hepatic insulin resistance and steatosis

To further confirm the role of BACH1 in HFD-induced insulin resistance, male eight-week-old loxP-*Bach1* transgenic mice were injected with AAV8-TBG-Cre to generate hepatocyte-specific *Bach1* overexpressed mice (*Bach1*[LTG], Supplementary Fig. 4a). Male eight-week-old loxP-*Bach1* transgenic mice injected with AAV8-TBG-GFP represented the controls (N[TG]) to *Bach1*[LTG] mice. Hepatic overexpression of *Bach1* in mice was validated by Western blot and immunohistochemistry, and BACH1 was specifically overexpressed in hepatocytes (Supplementary Fig. 4b–c). Two weeks after the AAV8 injection, mice were treated with CD or HFD for 12 weeks (Supplementary Fig. 5a). There were no significant differences in the body weight, food intake, levels of fasting blood glucose and fasting insulin, glucose tolerance, and insulin tolerance between *Bach1*[LTG] mice and N[TG] control mice with CD feeding (Supplementary Fig. 5b–c and Fig. 3a–d). Compared with the N[TG] control mice, *Bach1*[LTG] mice had similar body weight and food intake (Supplementary Fig. 5b–c) but increased liver/body weight ratios after HFD feeding (Fig. 3a). Similarly, HFD-fed *Bach1*[LTG] mice exhibited higher fasting blood glucose, fasting serum insulin levels, and HOMA-IR values than HFD-fed N[TG] mice (Fig. 3b). Furthermore, BACH1 overexpression in hepatocytes impaired glucose tolerance and decreased insulin tolerance according to the GTT (Fig. 3c) and ITT (Fig. 3d) results under HFD conditions. A significantly lower glycogen content was observed in the livers of HFD-fed *Bach1*[LTG] mice compared to HFD-fed N[TG] mice (Fig. 3e, 3f). Accordingly, the qRT-PCR assay showed an increase in the mRNA levels of gluconeogenesis genes *Pck1* and *G6pc* in the livers of HFD-fed *Bach1*[LTG] mice (Fig. 3g). The insulin-induced phosphorylation of IR-β, AKT, GSK-3β, and FOXO1 was notably decreased in the liver tissues of HFD-fed *Bach1*[LTG] mice as compared to HFD-fed N[TG] mice (Fig. 3h). Serum ALT, AST, and ALP levels, as well as TC, TG, and NEFA levels, showed a trend to increase in HFD-fed *Bach1*[LTG] mice compared to HFD-fed N[TG] mice (Supplementary Fig. 5d–i). Histological studies showed that *Bach1* overexpression aggravated HFD-induced hepatic steatosis (Supplementary Fig. 5j, k). Together, these data demonstrated that overexpression of hepatocytic *Bach1* aggravated HFD-induced hepatic insulin resistance and steatosis.

## BACH1 inhibits insulin signaling

To further elucidate the hepatocyte-specific mechanisms of the observed improved insulin signaling in *Bach1*[LKO] mice, we isolated primary hepatocytes from *Bach1*[LKO] and control mice and stimulated them with insulin. We found that BACH1 deficiency led to elevated phosphorylation levels of IR-β, AKT, GSK-3β, and FOXO1 in primary hepatocytes upon insulin stimulation compared with the control

hepatocytes (Fig. 4a). Similar results were observed in HepG2 cells when BACH1 was knocked down by infection with adenovirus interference vector of BACH1 (sh*Bach1*) (Supplementary Fig. 6a). The opposite effects were observed in the primary hepatocytes isolated from *Bach1*[LTG] mice (Fig. 4b) and HepG2 cells with BACH1 overexpression induced by the adenovirus vector of BACH1 (AdBACH1) (Supplementary Fig. 6b). We further performed these experiments at increasing concentrations of insulin (25-200 nM). The results showed that HepG2 cells infected with AdBACH1 had weakened and delayed phosphorylation of IR-β, AKT, GSK-3β, and FOXO1 at multiple insulin concentrations (Fig. 4c). Similar to the role of BACH1 in hepatocytes, our data showed that BACH1 also inhibited the insulin signaling in the skeletal muscle cell lines (C2C12) and murine pre-adipocytes (3T3-L1 cells), and the insulin signaling upon insulin stimulation was enhanced when BACH1 was knocked down in both cells (Supplementary Fig. 7a, b). Therefore, BACH1 blunted insulin signaling, at the level or upstream of IR-β autophosphorylation.

Lipophagy, a type of selective autophagy, could specifically degrade the excess lipid to regulate lipid storage in the cell[34]. To investigate whether lipophagy was affected by BACH1, we performed a colocalization study between the autophagosomal marker LC3 and lipid droplets (labeled by BODIPY). Our results showed that BACH1 had little effect on lipophagy (Supplementary Fig. 8a) and on the protein levels of lipophagy marker PLIN2 and autophagy makers, including LC3, P62, and LAMP1 in OA-treated HepG2 cells (Supplementary Fig. 8b).

## BACH1 interacts with PTP1B and IR-β by the BTB domain

BACH1 contains a BTB domain at the N-terminal region to interact with other proteins and a Bzip domain at the C-terminal region to bind DNA that regulates gene transcription[25]. To determine which specific regions of BACH1 affected the insulin signaling, we determined the protein levels of the insulin signaling molecules in BACH1 KO hepatocytes by expressing full-length BACH1, BACH1 lacking BTB, and BACH1 lacking Bzip. We found that BACH1 deficiency led to elevated phosphorylation levels of IR-β, AKT, and GSK-3β in primary hepatocytes isolated from *Bach1*[LKO] mice upon insulin stimulation, which were significantly inhibited by vectors coding for HA-tagged full-length BACH1 sequence (BACH1-HA) and for truncated sequences lacking the Bzip domain (BACH1[ΔBzip]-HA), but not by vectors coding for truncated sequences lacking the N-terminal BTB domain of BACH1 (BACH1[ΔBTB]-HA) (Fig. 5a). Consistently, the levels of insulin-stimulated phosphorylated IR-β, AKT, and GSK-3β in HepG2 cells were significantly inhibited by full-length BACH1 (BACH1-FLAG) and BACH1 lacking Bzip (BACH1[ΔBzip]-FLAG) vectors (Supplementary Fig. 9a), but not by BACH1 lacking BTB adenoviruses (AdBACH1[ΔBTB]) (Supplementary Fig. 9b). These results suggested that the BTB domain, but not the Bzip domain of BACH1 is essential for repressing the insulin signaling. Because heme oxygenase-1 (HO-1) expression is highly sensitive to the transcription factor activity of BACH1[35], we also determined the HO-1 expression in the above experiment. We found that HO-1 expression in BACH1 KO cells was significantly inhibited by full BACH1, but not by BACH1[ΔBzip] or BACH1[ΔBTB] (Fig. 5a), which indicated that both the BTB

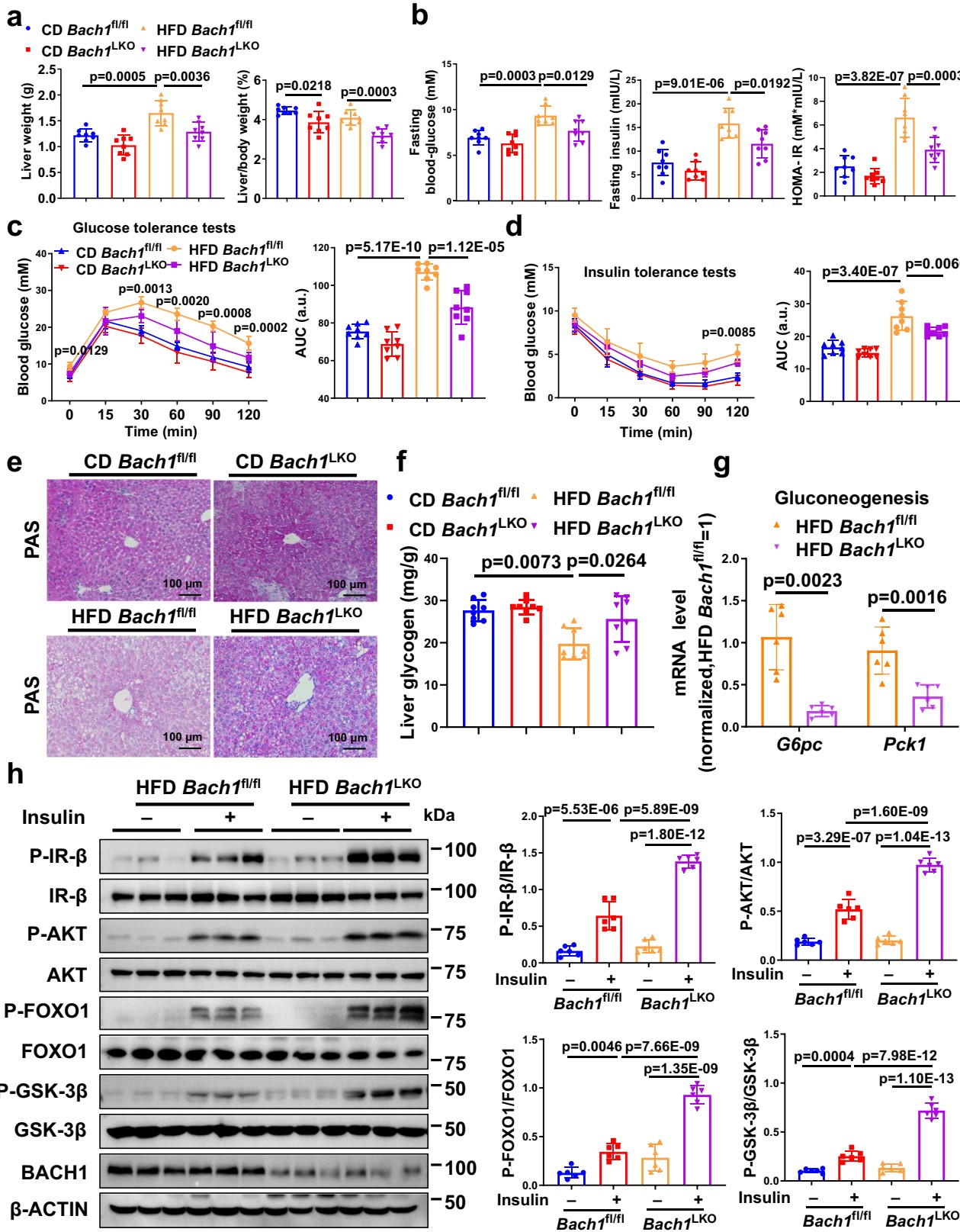

domain and the Bzip domain of BACH1 are important for the regulation of HO-1 in hepatocytes.

Given that insulin-induced IR-β phosphorylation is regulated by BACH1, we next asked whether BACH1 regulates insulin signaling via directly modulating the expression or interaction of IR-β or its regulators. The phosphatases serve as important negative regulators of insulin signaling, including PTP1B (dephosphorylates IR-β and IRS), serine/threonine protein phosphatase 2 A (PP2A) (dephosphorylates AKT and PI3K), and phosphatase and tensin homolog (PTEN) (dephosphorylates phosphatidylinositol 3,4,5-trisphosphate)[36]. By coimmunoprecipitation (co-IP) assay, we found that BACH1 interacted with IR-β and PTP1B, but not PTEN and PP2A

**Fig. 2 | Hepatic BACH1 deficiency improves obesity-induced insulin resistance and dysregulation of glucose homeostasis. a** Liver weight (left) and liver/body weight ratio(right) of male *Bach1*[LKO] and *Bach1*[fl/fl] mice fed a CD or HFD for 12 weeks were examined (*n* = 8 mice per group). *Bach1*[LKO]: hepatocyte-deficient *Bach1* mice, *Bach1*[fl/fl]: the controls to *Bach1*[LKO] mice. **b** Fasting blood glucose levels (left), fasting insulin levels (middle), and corresponding homeostasis model assessment of insulin resistance (HOMA-IR) index (right) of *Bach1*[LKO] and *Bach1*[fl/fl] mice after 12 weeks of CD or HFD treatment (*n* = 8 mice per group). **c** Glucose tolerance tests (GTT) were measured at week 12 of CD or HFD feeding (*n* = 8 mice per group). The area under the curve (AUC) was used to quantify the GTT results. **d** Insulin tolerance tests (ITT) were measured at week 12 of CD or HFD feeding (*n* = 8 mice per group). The AUC was used to quantify the ITT results. **e** Periodic acid-Schiff (PAS) staining of liver sections from *Bach1*[LKO] and *Bach1*[fl/fl] mice fed a CD or HFD for 12 weeks (scale bar = 100 μm, *n* = 8 mice per group). **f** A glycogen assay was carried out to determine the glycogen content in the livers of *Bach1*[LKO] and *Bach1*[fl/fl] mice (*n* = 8 mice per group). **g** mRNA expression of phosphoenolpyruvate carboxykinase1 (*Pck1*) and glucose-6-phosphatase catalytic subunit (*G6pc*) was measured in the livers of *Bach1*[LKO] and *Bach1*[fl/fl] mice with HFD feeding for 12 weeks by real-time quantitative PCR (qRT-PCR) (*n* = 6 mice per group). **h** Left: Western blot analysis of essential markers of insulin signaling (insulin receptor β (IR-β), AKT, forkhead box protein O1 (FOXO1), and glycogen synthase kinase-3β (GSK-3β)) in the liver tissues from *Bach1*[LKO] and *Bach1*[fl/fl] mice fed with HFD after injected i.p. with insulin after overnight fasting (*n* = 6 mice per group). Right: Phosphorylated protein levels were normalized to total protein. Statistical analysis was performed by two-way ANOVA followed by Tukey's test for (**a**–**d**), and (**h**), by two-way ANOVA followed by Kruskal–Wallis test with Dunn multiple comparisons test for (**f**), by two-tailed Welch test for *G6pc* in (**g**), by unpaired two-tailed Student's *t*-test for *Pck1* in (**g**). Data are presented as mean values ± SD. Source data are provided as a Source Data file.

(Fig. 5b). Overexpression of BACH1 in hepatocytes had no detectable influence on the expression of IR-β and PTP1B (Fig. 4b and Supplementary Fig. 11a). By using an immunofluorescence assay, we found that BACH1 was mainly detected in the cytoplasm, and enriched in the endoplasmic reticulum of HepG2 cells (Supplementary Fig. 10a). The deletion of the BTB domain of BACH1 resulted in its nuclear accumulation (Supplementary Fig. 10b), which is consistent with a previous study[22]. Because the activities of PTP1B have very high turn-over rates, a substrate-trapping mutant of PTP1B (PTP1B-D181A) was used. PTP1B-D181A retained the ability to bind to IR-β but could not dephosphorylate its substrates[37]. BACH1-GFP was co-transfected with PTP1B-D181A-BFP and IR-β-cherry in HepG2 cells. Immunofluorescence staining showed colocalization of the three proteins primarily to the cellular space surrounding the nucleus (Fig. 5c), which is consistent with a previous report about the distribution for PTP1B-D181A and IR-β[37]. Consistently, Glutathione s-transferase (GST) pull-down assay performed with the lysate from HEK293T cells demonstrated GST-tagged BACH1 interacted directly with IR-β-GFP (Fig. 5d) or PTP1B-HA (Fig. 5e) respectively. We further determined which specific regions of BACH1 interacted with PTP1B and IR-β by performing co-IP analyses in HEK293T cells that had been transfected with two vectors, one coding for PTP1B-HA or IR-β-GFP, and the other coding for Flag-tagged versions of the full-length BACH1 or BACH1[ΔBTB]. PTP1B or IR-β coimmunoprecipitated with the full-length BACH1 but not with BACH1[ΔBTB] (Fig. 5f, g). Thus, BACH1 binds directly with PTP1B and IR-β, and this interaction is mediated by the BTB domain in the N-terminal region of BACH1.

## BACH1 enhances the interaction between PTP1B and IR-β in response to insulin

To elucidate whether BACH1 affects the interaction between IR-β and PTP1B in response to insulin, we compared the interaction between IR-β and PTP1B in the liver tissues of HFD-fed *Bach1*[LKO] mice and HFD-fed control mice after insulin administration. Insulin stimulated the formation of the PTP1B-IR-β complex, which was significantly decreased in *Bach1*[LKO] mice livers compared with that in the control mice livers (Fig. 6a). These findings suggested that loss of BACH1 reduced the interaction between IR-β and PTP1B upon insulin stimulation. Consistent with these results, upon insulin stimulation, overexpression of BACH1 resulted in facilitating the accumulation of IR-β in the perinuclear compartment and enhancing the interaction of PTP1B-D181A and IR-β in the perinuclear compartment in HepG2 cells (Fig. 6b). However, the localization of BACH1 was not altered at 1, 5, 15 or 30 min after insulin stimulation as evidenced by immunofluorescent staining and BACH1 was not present on the cell membrane after insulin stimulation by Western blot analysis (Supplementary Fig. 10c, d). These findings indicated that BACH1 enhances the interaction between PTP1B and IR-β in response to insulin.

## BACH1 aggravates insulin resistance in a PTP1B-dependent manner

We then investigated whether PTP1B is involved in BACH1-regulated insulin signaling. Overexpression of BACH1 significantly impaired insulin-stimulated phosphorylation of IR-β, AKT, GSK-3β, and FOXO1 in primary hepatocytes, and this effect was rescued by knockdown of PTP1B (Fig. 6c) or PTP1B inhibitor (Supplementary Fig. 11b). To confirm the role of PTP1B in BACH1-mediated insulin resistance on HFD in vivo, we silenced PTP1B by administering AAV of the 8-serotype containing a construct encoding murine PTP1B (*Ptpn1* gene) short hairpin RNA (shRNA) under the control of TBG promoter for hepatocyte-specific expression (AAV-sh*Ptpn1*) to selectively inhibit PTP1B in hepatocytes in vivo. We injected *Bach1*[LTG] mice or N[TG] mice with AAV-sh*Ptpn1* and then fed mice with HFD for 12 weeks. Knockdown of PTP1B in hepatocytes did not affect body weight (Supplementary Fig. 12a), but significantly decreased the liver weight, liver/body weight, and the serum fasting blood glucose and insulin levels in *Bach1*[LTG] mice (Fig. 7a, b). Inhibition of PTP1B also counteracted the changes in the HOMA-IR index observed in *Bach1*[LTG] mice (Fig. 7b, right), improved glucose tolerance, insulin tolerance, and hepatic function, as well as reduced liver lipid accumulation in *Bach1*[LTG] mice (Fig. 7c, d, Supplementary Fig. 12b–i). Consistently, the decreases in glycogen content in the liver of *Bach1*[LTG] mice were partially abolished by PTP1B knockdown (Fig. 7e, f). Inhibition of PTP1B also abrogated the effect of BACH1 overexpression on insulin-stimulated phosphorylation of IR-β, AKT, GSK-3β, and FOXO1 (Fig. 7g). Considered together, these experiments demonstrated that the knockdown of PTP1B can restore insulin signaling in hepatocytes with BACH1 overexpression and block the decreased insulin signaling in *Bach1*[LTG] mice. Thus, BACH1-mediated aggravation of HFD-induced hepatic insulin resistance is mediated, at least in part, by PTP1B.

## Knockdown of BACH1 ameliorates hyperglycemia and insulin resistance in diabetic mice

To determine the functional role of hepatic BACH1 in the insulin resistance of diabetic mice, we silenced BACH1 by administering AAV-sh*Bach1* to selectively inhibit BACH1 in hepatocytes in *db/db* mice. We injected four-week-old male *db/db* mice with AAV-sh*Bach1* or AAV-sh*Con* to examine whether decreased BACH1 expression could reverse insulin resistance in *db/db* mice. AAV-sh*Bach1* infection caused efficient knockdown of endogenous BACH1 in the liver (Fig. 8i). After the AAV injection, *db/db* mice were treated with CD for four weeks. Knockdown of BACH1 in *db/db* mice had no significant effect on body weight and food intake (Supplementary Fig. 13a, b). Consistent with the effect of HFD-fed *Bach1*[LKO] mice, *db/db* mice injected with AAV-sh*Bach1* exhibited decreased fasting blood glucose, improved glucose tolerance, insulin tolerance, and improved hepatic insulin signaling (Fig. 8a–c, i, and Supplementary Fig. 13c). To measure the effect of BACH1 deficiency on insulin sensitivity more accurately, we performed

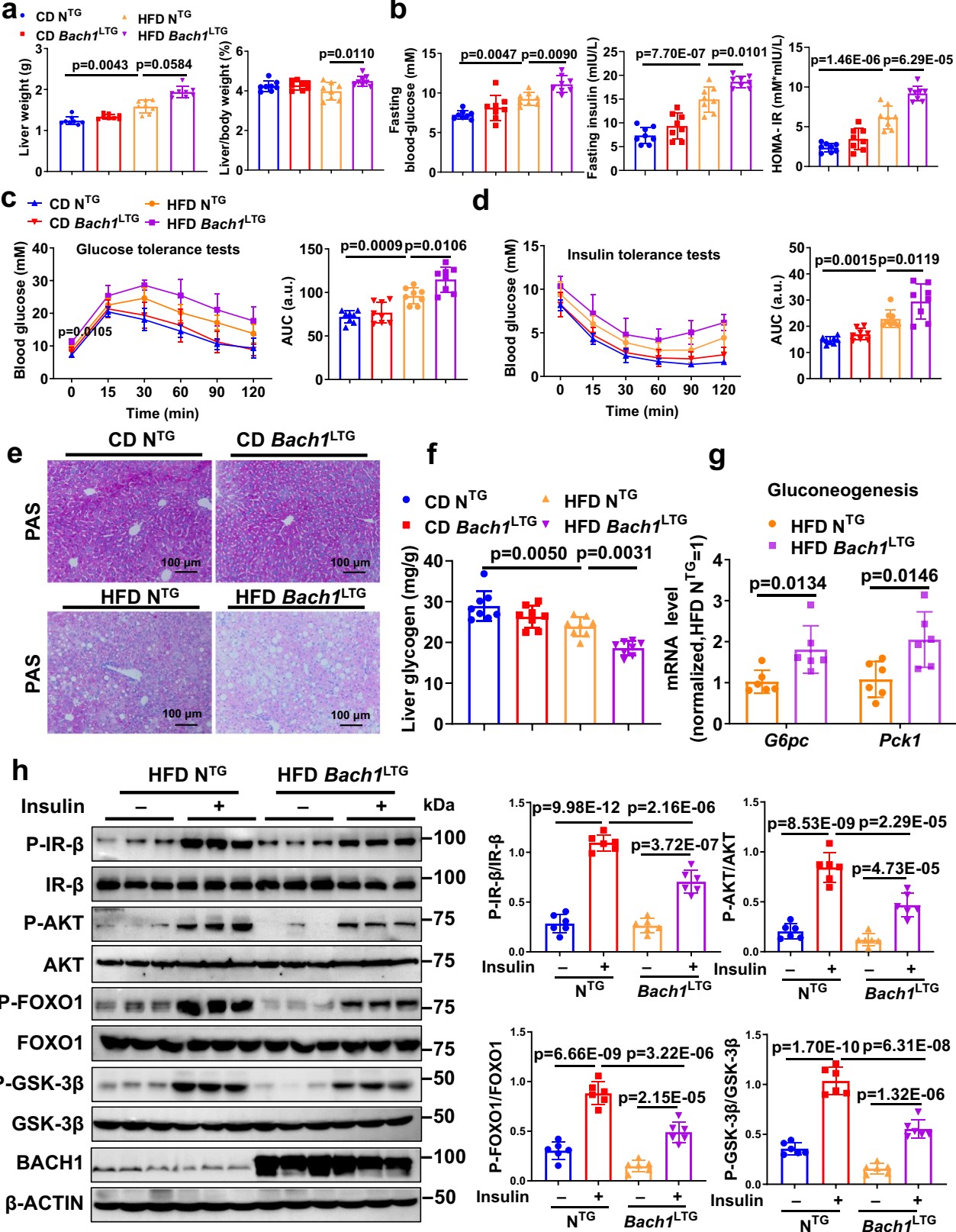

hyperinsulinemic/euglycemic clamp studies in *db/db* mice. The glucose infusion rate (GIR), as quantified by the amount of exogenous glucose required to maintain euglycemia, was higher in AAV-sh*Bach1* *db/db* mice than in the control *db/db* mice (Fig. 8d). The insulin-stimulated suppression of hepatic glucose production (HGP) was also significantly enhanced in AAV-sh*Bach1* mice when compared to the control mice (Fig. 8e, f). Moreover, the knockdown of BACH1 in *db/db* mice enhanced the hepatocyte glycogen content and reduced the expression of gluconeogenesis-related genes in the livers of mice (Fig. 8g, h). Thus, the hepatic-specific knockdown of BACH1 appeared to reduce blood glucose levels by both increasing insulin sensitivity and reducing hepatic glucose production.

**Fig. 3 | Hepatic overexpression of BACH1 facilitates HFD-induced insulin resistance. a** Liver weight (left) and liver/body weight ratio (right) of male $Bach1^{LTG}$ and $N^{TG}$ mice fed a CD or HFD for 12 weeks (n = 8 mice per group). $Bach1^{LTG}$: hepatocyte-specific $Bach1$ overexpressed mice, $N^{TG}$: the controls to $Bach1^{LTG}$ mice. **b** Fasting blood glucose levels (left), fasting insulin levels (middle), and HOMA-IR index (right) of $Bach1^{LTG}$ and $N^{TG}$ mice after 12 weeks of CD or HFD treatment (n = 8 mice per group). **c** GTTs were measured at week 12 of CD or HFD feeding (n = 8 mice per group). The area under the curve (AUC) was used to quantify the GTT results. **d** ITTs were measured at week 12 of CD or HFD feeding (n = 8 mice per group). The AUC was used to quantify the ITT results. **e** PAS staining of liver sections from $Bach1^{LTG}$ and $N^{TG}$ mice fed a CD or HFD for 12 weeks (scale bar = 100 μm, n = 8 mice per group). **f** A glycogen assay was carried out to determine the glycogen content in the liver tissues of $Bach1^{LTG}$ and $N^{TG}$ mice (n = 8 mice per group). **g** mRNA expression of $Pck1$ and $G6pc$ was measured in the liver tissues of $Bach1^{LTG}$ and $N^{TG}$ mice with HFD feeding for 12 weeks by qRT-PCR (n = 6 mice per group). **h** Left: Western blot analysis of essential markers of insulin signaling (IR-β, AKT, FOXO1, and GSK-3β) in the liver samples from $Bach1^{LTG}$ and $N^{TG}$ mice fed with HFD after injected i.p. with insulin after overnight fasting (n = 6 mice per group). Right: Phosphorylated protein levels were normalized to total protein. Statistical analysis was performed by two-way ANOVA followed by Kruskal−Wallis test with Dunn multiple comparisons test for a (left), by two-way ANOVA followed by Tukey post hoc tests for (**a**) (right), (**b**–**d**), (**f**), and (**h**), by two-tailed Student's t-test for (**g**). Data are presented as mean values ± SD. Source data are provided as a Source Data file.

## Discussion

The molecular mechanisms responsible for hepatic insulin resistance in metabolic diseases such as obesity or diabetes are not completely clear. Here, we showed that hepatocyte-specific loss of $Bach1$ improved insulin signaling and dysregulation of glucose homeostasis in HFD-induced hepatic insulin resistance, and consequently protected from HFD-induced steatosis, whereas hepatic overexpression of $Bach1$ in mice led to the opposite phenotype, indicating that BACH1 is a negative regulator of insulin signaling. Mechanistically, BACH1 directly interacted with PTP1B and IR-β in hepatocytes, and loss of hepatic BACH1 reduced the interaction between PTP1B and IR-β upon insulin stimulation in HFD-fed mice, thus improving insulin resistance and glucose metabolism (Fig. 8j). Moreover, BACH1 knockdown significantly ameliorated hyperglycemia and insulin resistance in the diabetic mouse models. These observations uncover the role of BACH1 in hepatocytes as a crucial regulator of insulin signaling and glucose metabolism.

Previous studies have shown that BACH1 expression was significantly increased in hypertrophic human hearts, human carotid and coronary atherosclerotic plaques, and tumors from patients with triple-negative breast cancer and human epithelial ovarian cancer[16,24,25,38–40]. In the present study, we observed that BACH1 expression was higher in the hepatocytes of individuals with obesity than those of lean individuals and in the livers of patients with NAFLD and diabetic mice than in normal control livers. The treatment of OA and PA upregulated the protein levels of BACH1 in primary hepatocytes. These results indicated that hepatic BACH1 might be upregulated by free fatty acids (FFAs). In line with this notion, a high-fat and high-fructose diet increased BACH1 expression in the livers of mice[30]. Significant upregulation of BACH1 was also observed in human islets exposed to palmitate, as well as human glomerular endothelial cells and the aorta of diabetic rats upon high glucose[27,41,42]. The reasons for the increase of BACH1 in response to OA or PA have yet to be determined. One possibility is that HFD may downregulate the expression of let-7a-5p microRNA[43], which is an inhibitor of BACH1[44]. Future work will be required to investigate the association of the BACH1 gene with genetic variants linked with the risk of insulin resistance, obesity, and diabetes.

Glycogen homeostasis includes the cooperative regulation of the rate of glycogen synthesis and the rate of glycogenolysis. Defects in these two processes might be the major causes of hyperglycemia in obesity and type 2 diabetes[45]. Consistent with increased hepatic insulin signaling in HFD-fed $Bach1^{LKO}$ mice, we observed that liver glycogen content was significantly increased and the expression of gluconeogenic genes ($Pck1$ and $G6pc$) was reduced in the livers of $Bach1^{LKO}$ mice. These data suggested that hepatocyte knockout of BACH1 in mice could suppress hyperglycemia most likely by promoting the conversion of blood glucose into liver glycogen and inhibiting glycogenolysis. Recently, it was reported that BACH2 inhibition reversed β cell failure in type 2 diabetes mice, and treatment with a BACH inhibitor lowered glycemia and increased plasma insulin levels in diabetic mice, and restored insulin secretion in diabetic mice and human islets[46], suggesting that BACH inhibition may develop as a promising therapeutic strategy for diabetes.

Increased expression of PTP1B and IR-β/PTP1B interaction has been reported in subjects with insulin resistance and obesity[47,48]. However, the molecular mechanism of the desensitization of IR-β signaling by PTP1B in an insulin resistance state remains elusive. Our results showed that under physiological conditions, BACH1 interacted with PTP1B and IR-β in the region surrounding the cell nucleus (primarily in ER), and the knockdown of BACH1 facilitated insulin-stimulated phosphorylation of IR-β and downstream signaling, indicating that BACH1 is a negative regulator of insulin signaling. Under pathological conditions with HFD feeding, the increased BACH1 expression induced by excessive nutrient intake impaired insulin signaling by facilitating the binding of PTP1B and IR-β in the perinuclear compartment. Accordingly, BACH1 deficiency in mice reduced the insulin-stimulated formation of the PTP1B and IR-β. Indeed, we observed that inhibition of PTP1B attenuated BACH1-mediated suppression of insulin signaling both in vivo and in vitro, suggesting that BACH1 controls hepatic insulin signaling, at least partially, in a PTP1B-dependent manner.

We found that deletion of the BTB domain of BACH1 also affected HO-1 activity as a transcription factor. Hepatocyte conditional HO-1 deletion in mice evoked resistance to diet-induced insulin resistance and reduced diet-induced fatty liver disease[49]. $Bach1$ ablation exerted a hepatoprotective effect against steatohepatitis presumably via HO-1 induction[29]. HO-1-dependent anti-oxidative effects might contribute to the observed protective role of $Bach1^{LKO}$ mice. Therefore, transcription regulation may also be involved in the function of BACH1 in regulating insulin signaling.

We have noticed that hepatic $Bach1$-overexpressing mice showed no significant changes in the body weight, levels of blood glucose, and insulin signaling on CD, although BACH1 overexpression inhibited insulin signaling in hepatocytes. These data suggested that overexpression of BACH1 alone under normal physiological conditions is not sufficient to produce a phenotype in mice and there may be other counterregulatory mechanisms at play. Instead, HFD-fed hepatic BACH1-overexpressing mice exhibited aggravated hepatic insulin resistance and steatosis. Moreover, the metabolic characteristics of our $Bach1^{LKO}$ mice are similar to those of liver-specific $Ptpn1$ knockout mice on HFD[50]. The BACH1 BTB domain may play an important role in the interaction of PTP1B and IR-β, and insulin signaling in hepatocytes. Previous evidence showed that the adapter protein NCK attenuated insulin signaling by recruiting PTP1B to the IR-β[51], similar to what was described for BACH1. Additional work is required to determine whether the inhibitor that targets the interaction between BACH1 and PTP1B is an alternative more specific approach for the treatment of diabetes.

A previous study showed that $Bach1$ deficiency in mice minimally impacted obesity and insulin resistance after high-fat diet loading[28], which contrasts with our results. These discrepant observations are unclear but might be related to the different compositions of the HFDs used (the feed we used contains more fat than theirs). Furthermore,

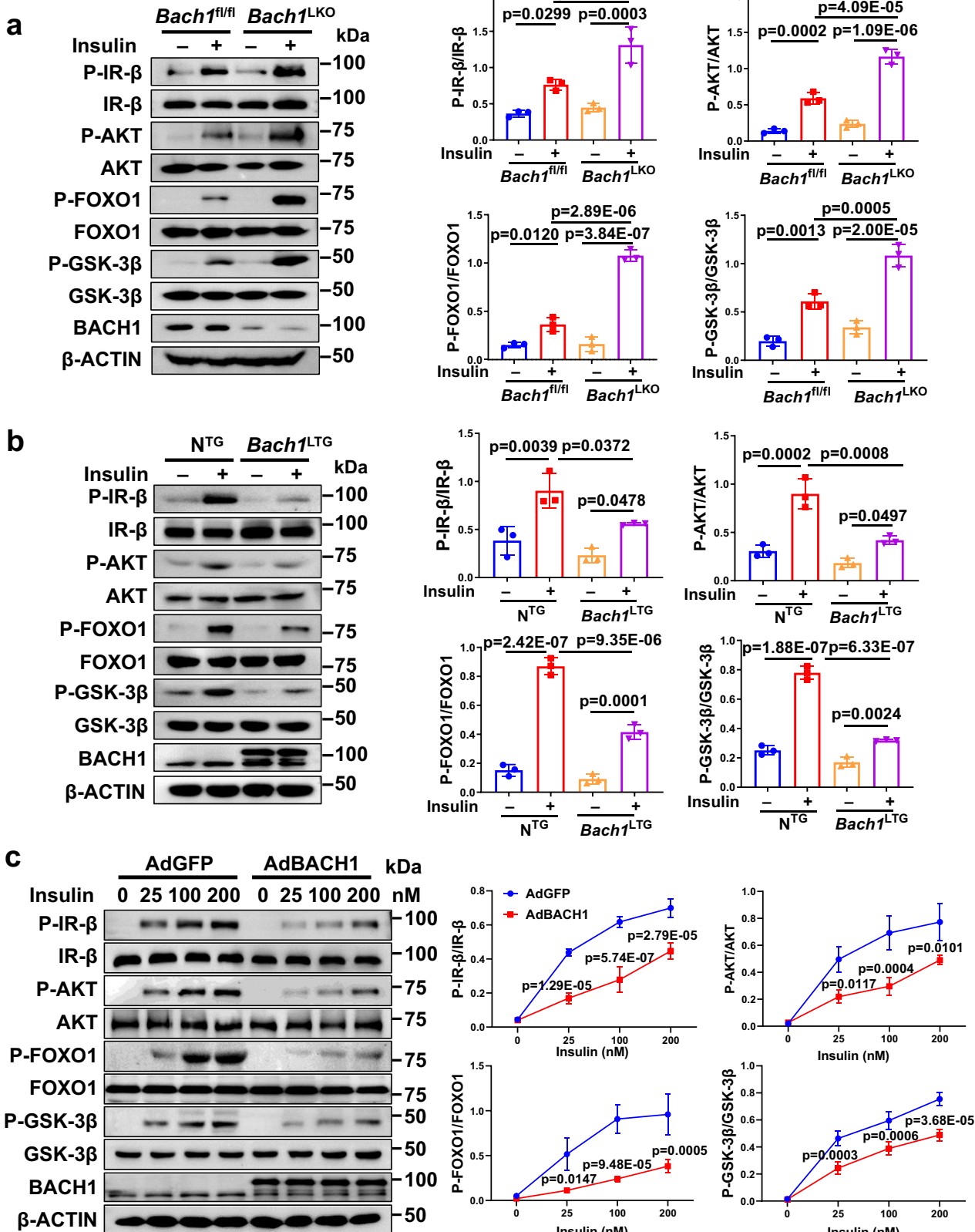

the study performed by Kondo et al. was conducted in global *Bach1* knockout mice[28]. BACH1 has been shown to increase glucose uptake and glycolysis in lung cancer cells[17], which is inconsistent with the results in our study. Thus, BACH1 may exert different actions in normal compared to tumor cells. Further, differential roles played by *Bach1* in

cancer cells and hepatocytes may be influenced by the cell types, the external environment, and the metabolic state.

Insulin resistance is strongly associated with hepatic steatosis. The improvement in insulin resistance via inhibition of sterol regulatory element binding protein-1c and fatty acid synthase ameliorates

**Fig. 4 | BACH1 inhibits insulin signaling in hepatocytes. a** Left: Western blot analysis of the phosphorylation and total protein levels of IR-β, AKT, FOXO1, and GSK-3β in the primary hepatocytes isolated from male *Bach1*[fl/fl] and *Bach1*[LKO] mice with or without insulin (100 nM) stimulation. Right: The phosphorylated protein levels normalized to total protein (*n* = 3 mice per group). **b** Left: Western blot analysis of the phosphorylation and total protein levels of IR-β, AKT, FOXO1, and GSK-3β in the primary hepatocytes isolated from N[TG] and *Bach1*[LTG] mice with or without insulin (100 nM) stimulation. Right: The phosphorylated protein levels normalized to total protein (*n* = 3 mice per group). **c** Left: Phosphorylation of key molecules of the insulin pathway was determined in the HepG2 cells infected with adenoviruses coding for BACH1 (AdBACH1) or GFP (AdGFP) and then stimulated with different concentrations of insulin for 10 min. Right: The phosphorylated protein levels normalized to total protein (*n* = 3 biological replicates). Statistical analysis was performed by two-way ANOVA followed by Tukey post hoc tests for (**a**–**c**). Data are presented as mean values ± SD. Source data are provided as a Source Data file.

hepatic steatosis[52]. We observed that the hepatocyte-specific deletion of BACH1 protected from HFD-induced steatosis, which may be associated with the improvement in insulin resistance in the liver. Modulation of liver lipophagy has the potential to ameliorate NAFLD, non-alcoholic steatohepatitis, and insulin resistance[53]. However, our results showed that autophagic lipid droplets degradation was not affected by BACH1, suggesting that hepatocyte-specific deletion of *Bach1* attenuated the lipid accumulation in the liver in a lipophagy-independent manner.

There are limitations in this work. Firstly, the mouse experiments were performed only in male animals. Secondly, we did not investigate the interaction of BACH1, PTP1B, and IR-β in animal models. Thirdly, intraperitoneal GTT may cause a poor insulin secretory response compared to oral gavage. Furthermore, the tolerance tests were dosed by body weight rather than lean mass, which may confound interpretation as obese animals will have a higher delivery of glucose or insulin than lean animals[54].

Overall, our findings demonstrated that BACH1 played a critical role in hepatic insulin signaling and glucose metabolism. BACH1 facilitated PTP1B binding to IR-β, suppressed insulin signaling, and aggravated insulin resistance in HFD-fed mice and diabetic mice. These findings suggested that BACH1 could be an attractive therapeutic target for the treatment of diabetes, potentially facilitating improved insulin sensitivity and glucose tolerance.

## Methods
### Ethics statement
All procedures involving human specimens in this study were approved by the Ethics Committee Board at the School of Basic Medical Sciences, Fudan University (approval number: 2016-002). Each patient provided written informed consent. All procedures complied with the ethical guidelines of the Declaration of Helsinki.

All animal handling protocols were approved by the Animal Care and Use Committee of the School of Basic Medical Sciences of Fudan University (20211021-004) and were conducted in accordance with the National Institutes of Health Guide for the Care and Use of Laboratory Animals.

### Animals and treatment
The *Bach1* loss-of-function mouse and the loxP-*Bach1* transgenic mouse in C57/BL background were generated by the Nanjing Biomedical Research Institute of Nanjing University. *Bach1*[flox/flox] mice were generated by flanking exon 3 and exon 4 of *Bach1* with loxP sites[24]. The heterozygous *Bach1* loxP mice were intercrossed to obtain homozygous *Bach1* loxP mice. Homozygous male eight-week-old *Bach1* loxP mice (*Bach1*[flox/flox]) (20 g) (*n* = 8) were injected with adeno-associated virus 8 (AAV8) expressing mouse thyroxine-binding globulin (TBG) promoter-driven Cre recombinase (AAV8-TBG-Cre), which were obtained from the Vigene Biosciences (Shandong, China), to generate *Bach1*[LKO] mice for hepatocyte-specific deletion of *Bach1*. A loxP-*Bach1* transgenic mouse was genetically engineered by inserting a single copy of the mouse *Bach1* CDS into the mouse H11 locus under a CAG promoter and a loxP-STOP-loxP cassette. Male eight-week-old loxP-*Bach1* transgenic mice (20 g) (*n* = 8) were injected with AAV8-TBG-Cre to generate hepatic *Bach1* overexpression (*Bach1*[LTG]) mice. Animals

from the same litter were randomly assigned to experimental groups after genotyping. Randomization and blinding were adopted for animal studies. After administering a single tail vein injection of AAV8-TBG-Cre with $1 \times 10^{11}$ viral particles in a 100 μl volume of sterile PBS, mice were fed with 12 weeks with HFD fodder (20% kcal protein, 60 kcal % fat and 20% kcal carbohydrate) from Shuyishuer Biotechnology Co. Ltd (#D12492, Changzhou, China,) or CD from Shanghai Laboratory Research Center (Shanghai, China). To measure food intake, we placed each mouse in a separate cage and weighed the chow once before feeding and again 24 h later. For the rescue experiment, we silenced *Ptpn1* by administering an AAV containing a construct encoding murine *Ptpn1* short hairpin RNA (shRNA) under the TBG promoter for hepatocyte-specific expression (AAV-sh*Ptpn1*, designed and synthesized by Gene Pharma, Shanghai, China). The sequences of shRNA are listed in Supplemental Table S1. After administering a single tail vein injection of AAV-sh*Ptpn1* with $1 \times 10^{11}$ viral particles in a 100 MI volume of sterile PBS, mice were fed for 12 weeks with HFD. At the end of treatment, all mice were fasted overnight and euthanized with pentobarbital (60 mg/kg, intraperitoneal injection) followed by cervical dislocation, then liver tissues and blood were collected. For in vivo analysis of insulin signaling, one set of mice in each group was fasted for four hours and received insulin (0.75 IU/kg) via intraperitoneal injection for ten minutes before the liver tissues were harvested. The primers used for mice genotyping are listed in Supplemental Table S2. Liver tissues from male eight-week-old WT and leptin-mutated (*ob/ob*) mice were from Dongning Pan's lab (Fudan University, China). Male four-week-old WT mice (15 g) or leptin receptor-mutated (*db/db*) mice (30 g) were obtained from Gem Pharmatech Co. Ltd (Nanjing, China). To generate hepatocyte-specific *Bach1* knockdown in *db/db* mice, four-week-old male *db/db* mice were injected with AAV8 containing a construct encoding murine *Bach1* shRNA under the TBG promoter for hepatocyte-specific expression (AAV-sh*Bach1*). After the AAV injection, *db/db* mice were treated with CD for four weeks, and then fasting blood glucose and other indicators were measured. The analyses of RNAseq data of the mice liver tissue were performed by R Software (4.0.2).

Mice were housed in standard cages with an SPF environment with a 12-h light/dark cycle at a room temperature of 22 °C ± 2 °C, humidity of 50% ± 5%, with free access to food and water. All animal studies and data analyses were performed by two independent investigators in a blinded manner.

### Human liver samples
Human liver samples (*n* = 6) were obtained painlessly from adult patients with NAFLD who underwent liver transplantation or liver biopsy at Zhongshan Hospital, Fudan University. The corresponding control liver tissues (*n* = 6) were harvested from donors whose livers were excluded from liver transplantation for non-hepatic reasons. The information including age, gender, and the degree of steatosis in the human liver samples are in Table S3. H&E, PAS, and immunohistochemistry staining were performed in liver tissue samples. An equal number of male and female human participants were used in our study. The gender of participants was determined based on self-report. The analyses of RNAseq data of the human liver tissue were performed by R Software (4.0.2).

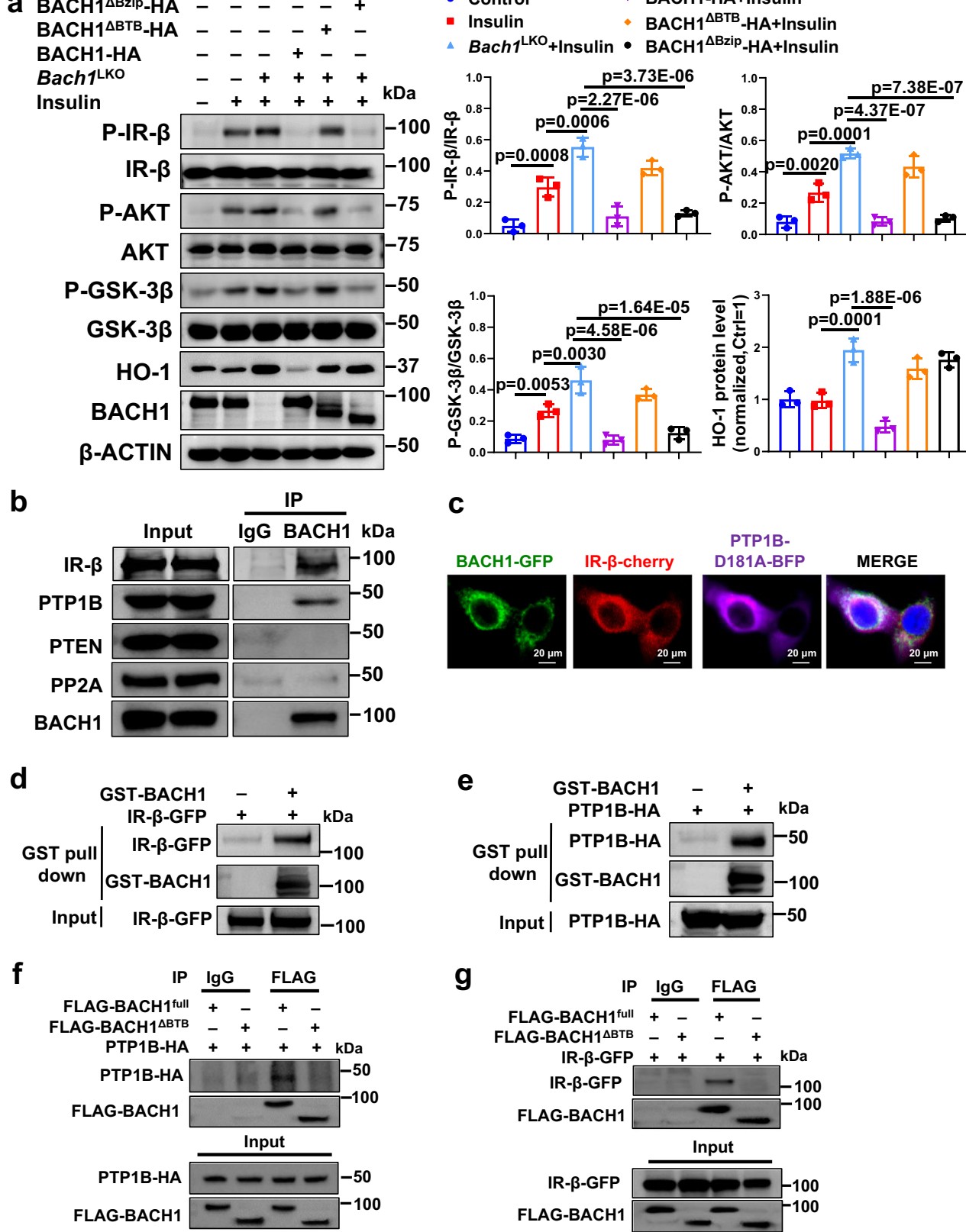

## Metabolic indicators and hepatic function detection

To perform glucose and insulin tolerance tests (GTT & ITT), male mice were fasted for 8 h or 4 h to confirm the correction of physiological response. Mice were given an intraperitoneal injection of glucose (2 g/ kg body weight) (# D810588, Macklin Inc., Shanghai, China). Then, the concentration of blood glucose of tail venous blood at 0 min, 15 min,

30 min, 60 min, and 120 min after glucose treatment were examined using commercial blood glucose test strips (ACCU-CHEK®, Roche Diabetes Care GmbH, Shanghai, China). For ITT, mice were treated with an intraperitoneal injection of insulin (1 U/kg body weight, Sigma–Aldrich). Subsequently, blood samples were collected from the tail vein at 0 min, 15 min, 30 min, 60 min, and 120 min post-injection

**Fig. 5 | BACH1 interacts with protein tyrosine phosphatase 1B (PTP1B) and IR-β by the BTB domain. a** Left: Western blot analysis of the phosphorylation and total protein levels of IR-β, AKT, and GSK-3β in primary hepatocytes isolated from *Bach1*[LKO] mice. Hepatocytes were transfected with vectors coding for HA-tagged versions of the full-length mouse *Bach1* sequence (BACH1-HA) or mutant sequences lacking the Bzip domain (BACH1[ΔBzip]-HA) or BTB domain (BACH1[ΔBTB]-HA) with or without insulin (100 nM) stimulation for 10 min. Right: Phosphorylated protein levels were normalized to total protein (*n* = 3 biological replicates). **b** The coimmunoprecipitation (co-IP) assay of BACH1 and IR-β, PTP1B, phosphatase and tensin homolog (PTEN), and serine/threonine protein phosphatase 2 A (PP2A) in mouse primary hepatocytes. Cell lysates were immunoprecipitated with control mouse IgG or BACH1, and immunoblotting was used to detect IR-β, PTP1B, PTEN, PP2A, and BACH1 (*n* = 3 biological replicates). **c** Representative immunostaining of BACH1-GFP (green), IR-β-cherry (red), and PTP1B-D181A-BFP (purple) in HepG2

cells. Nuclei were stained with DAPI (blue) (Scale bar = 20 μm, *n* = 3 biological replicates). **d, e** Glutathione s-transferase (GST) pull-down assay. The lysate from HEK293T cells expressed IR-β-GFP (**d**) or expressed PTP1B-HA (**e**) were incubated with GST-tagged BACH1 protein (GST-BACH1). Immunoblotting was used to detect IR-β-GFP and GST-BACH1 (**d**) or PTP1B-HA and GST-BACH1 (**e**) (*n* = 3 biological replicates). **f, g** The co-IP analyses in HEK293T cells. HEK293T cells were transfected with two vectors, one coding for PTP1B-HA (**f**) or IR-β-GFP (**g**), and the other coding for Flag-tagged versions of the full *Bach1* sequence or deletion sequences lacking the BTB domain. The BACH1 was immunoprecipitated with an anti-Flag antibody. PTP1B was detected in the precipitate with an anti-HA antibody (**f**), and IR-β was detected in the precipitate with an anti-GFP antibody (**g**) (*n* = 3 biological replicates). Statistical analysis was performed by one-way ANOVA followed by Tukey post hoc tests for a. Data are presented as mean values ± SD. Source data are provided as a Source Data file.

for measurement of glucose levels[55,56]. The HOMA-IR index was calculated from fasting levels of glucose and insulin in serum, respectively. Hepatic glycogen was extracted and determined using a glycogen assay kit (#A043-1-1, Nanjing Jiancheng Bioengineering Institute, Nanjing, China). The levels of serum aspartate aminotransferase (AST) (#C010-3), alanine aminotransferase (ALT) (#C009-3), alkaline phosphatase (ALP) (#A059-1), triglycerides (TG) (#A110-2), total cholesterol (TC) (#A111-2) and non-esterified fatty acids (NEFA) (#A042-1) (Nanjing Jiancheng Bioengineering Institute, Nanjing, China), as well as serum insulin (#ab277390, Abcam, Cambridge, UK) were detected using commercially-available detection kits.

### Hyperinsulinemic-euglycemic clamping

For catheter implantation surgery, *db/db* mice underwent surgical catheterization of the jugular vein under isoflurane anesthesia. Mice were given a single post-operative dose of carprofen (2.5 mg/kg) and buprenorphine (0.05 mg/kg) (slow release, sub-cutaneous) before surgery and after surgery. The left carotid artery (for blood sampling) and the right jugular vein (for glucose solution and insulin infusion) were catheterized. 2–3 days after surgery, the mice were fasted for 5 h (starting at 08:00 a.m.), kept awake, and allowed to move freely during the test. The clamp procedure consisted of a 120-min tracer equilibration period, followed by a 120-min clamp period. At the beginning of the equilibration period, the *db/db* mice were infused with 3-[3H]-D-glucose as a bolus dose (5 μCi) for 2 min and then decreased to a rate of 0.05 μCi/min for 2 h. Human insulin was continuously infused at 1.2 mU/kg/min (0–120 min), and the [3-3H]-glucose infusion was adjusted to 0.1 μCi/min to maintain euglycemia (90 mg/dL). All *db/db* mice blood was collected from the tip of the tail in heparinized capillary tubes and immediately spun down to collect plasma for tracer and insulin analysis. Glucose was assessed from whole blood directly from the tail tip with a Bayer Contour Blood Glucose Meter every 10 min for 120 min. The glucose infusion rate was adjusted to maintain euglycemia (100 mg/dL). At the end of the clamp experiment, mice were sacrificed, and the livers were snap-frozen in liquid nitrogen for measurements of glycogen content. The insulin infusion rate was based on observations in pilot studies.

### Reagents

The main reagents used in this paper are listed as follows: DMEM (Life Technologies, Carlsbad, CA, USA, #12800082); fetal bovine serum (Gbico, Carlsbad, CA, USA, #10099-141); bovine serum albumin (Yeasen, Shanghai, China, #36101ES25); PMSF (Beyotime, Shanghai, China, # ST505); Protease inhibitor cocktail (MedCheExpress, Shanghai, China, #HY-K0010); DAPI (Sigma–Aldrich, St. Louis, MO, USA, #D9542-1MG); Triton X-100 (Sigma–Aldrich, St. Louis, MO, USA, #T9284); Formaldehyde solution (Sigma–Aldrich, St. Louis, MO, USA, #F8775); Hipure Gel Pure DNA Mini Kit (Magen, Guangzhou, China, #D2111-02); Protein A/G PLUS-Agarose (Santa Cruz, Santa Cruz, CA, USA, #SC-2003); ReverTra Ace qPCR RT kit (Toyobo, Osaka, Japan, #FSQ-101);

Lipofectamine™ 3000 Transfection Reagent (Gbico, Carlsbad, CA, USA, #L3000075); Opti-MEM (Gbico, Carlsbad, CA, USA, #31985088); Trizol reagent (Invitrogen, Carlsbad, CA, USA, #15596026); qPCR SYBR® Green Master Mix (Toyobo, Osaka, Japan, #QPK-201).

### RNA extraction and qRT-PCR for gene expression

Total RNA was collected from liver tissues using Trizol Lysis Reagent (Invitrogen, Carlsbad, CA, USA, #15596026). RNA was converted to cDNA using a High-Capacity cDNA Reverse Transcription Kit from Toyobo. PCR was performed with a Bio-Rad iQ5 real-time PCR thermal cycler, using SYBR Green Supermix (Toyobo, Osaka, Japan, #QPK-201). The program used for amplification was 40 cycles of 15 s at 95 °C, 30 s at 58 °C, and 30 s at 72 °C, preceded by 5 min at 95 °C or enzyme activation. The $2^{-\Delta\Delta CT}$ method was used to analyze the relative changes in gene expression from qRT-PCR experiments. Relative gene expression to that of housekeeping gene β-ACTIN was calculated. Data were normalized and the control group was set at 1.0[24]. Primers are listed in Supplemental Table S4. Primer design was performed by Primer Premier 5.0 software.

### Histological examination

To show lipid deposition in the liver, liver tissues were embedded in Tissue-Tek OCT, frozen, sectioned, and then stained with Oil Red O. To perform histologic and immunohistochemical analysis, the liver tissue samples were accordingly fixed with 4% paraformaldehyde (Solarbio Life Sciences, Beijing, China, #P1110), embedded in paraffin, and sectioned transversely. The thin sections of liver samples were stained with H&E to observe the pattern of impaired status of tissues. To visualize glycogen levels in the liver, sections were stained with PAS. For the immunohistochemistry assay, the tissue sections were immersed in sodium citrate buffer (10 mM, pH 6.0) and heat retrieved for 20 min in a 100 °C water bath for antigen retrieval. The slices were permeabilized and blocked in PBS-T (0.03% Triton X-100) with 1% goat serum for 1 h at RT. Human liver sections were incubated with BACH1 antibody (Proteintech, Rosemont, IL, USA, #14018-1-AP, dilution 1:100) overnight at 4 °C, followed by incubation with horseradish peroxidase-conjugated goat anti-rabbit-IgG (Thermo Fisher Scientific, Shanghai, China, 31430, dilution 1:100). Normal rabbit (Cell Signaling Technology, Danvers, MA, USA, #3900, dilution 1:100) IgG as histology controls. Immunohistochemical staining was visualized using DAB (Zhongshan Biotech, Beijing, China, #ZLI-9032). Images were captured with a Leica DMI6000B microscope (Leica Microsystems, Buffalo Grove, IL, USA). The immunohistochemical quantification of human liver samples was performed by NIH Image J software[57]. The minimum threshold value was set at zero. The maximum threshold value was set so that the background signal was removed without removing the true signal. After setting the threshold, the image was converted to a black-and-white image. The mean gray value represents the quantified intensity. The relative protein expression value was normalized by the intensity value. Immunohistochemical quantification of mouse livers

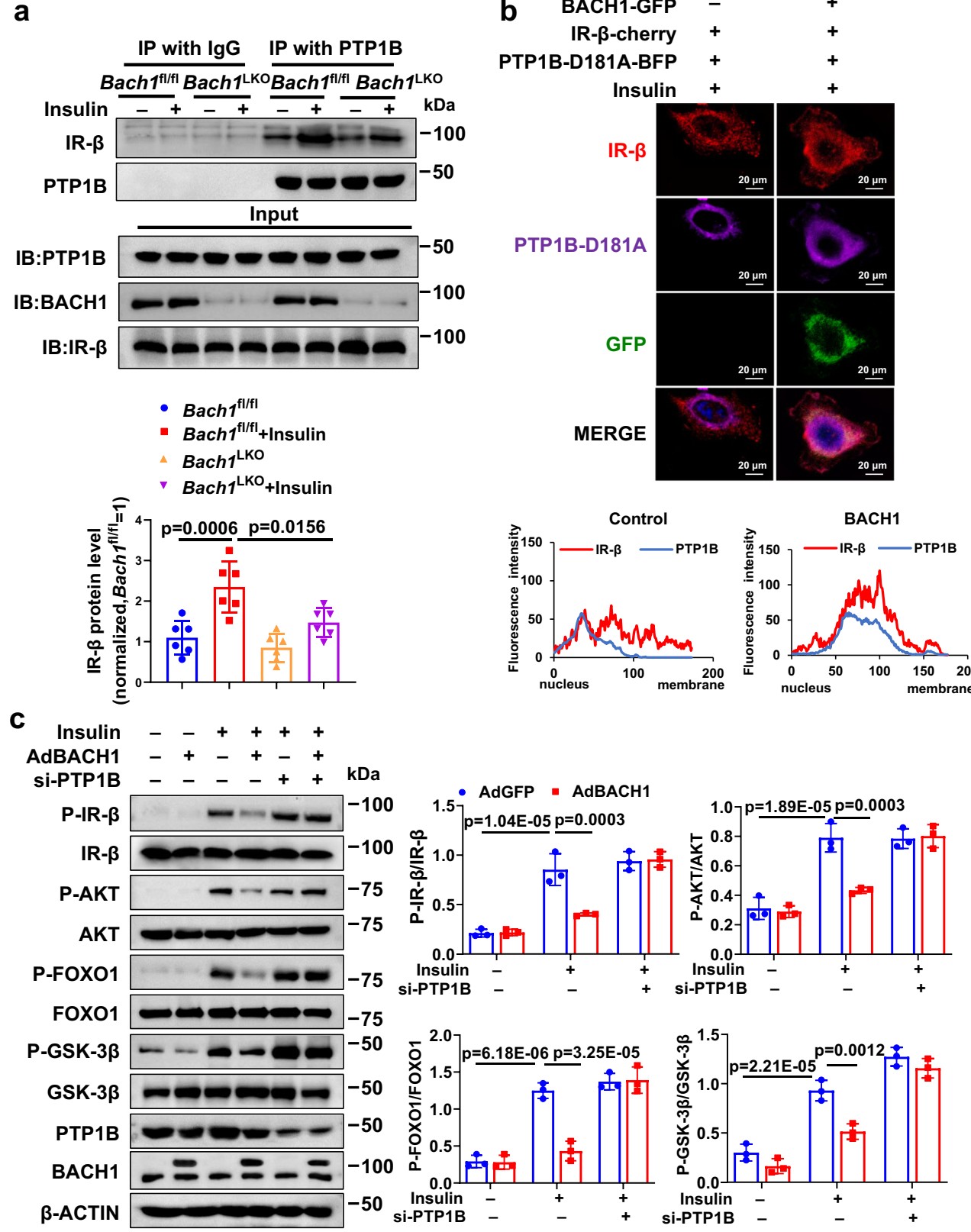

was performed by counting the number of positive stained cells in 10 randomly selected non-overlapping fields under a 40× magnification. 3 tissue sections from different animals of each group were used for statistical analysis. Representative images were chosen to reflect the mean value of quantitative data.

## Cell culture and treatment

Primary hepatocytes were isolated from 6 to 8-week-old male mice[58]. Hepatocytes were cultured in Dulbecco's modified Eagle's medium containing 10% fetal bovine serum and 1% penicillin-streptomycin. HepG2 hepatocyte (Cat# TCHu 72), myoblast C2C12 (Cat# SCSP-505),

**Fig. 6 | BACH1 enhances the interaction between PTP1B and IR-β in response to insulin. a** The co-IP analyses of the interaction of IR-β and PTP1B in the liver lysates from HFD-fed male *Bach1*<sup>LKO</sup> and *Bach1*<sup>fl/fl</sup> mice. Mice were injected i.p. with insulin (100 nM) after overnight fasting. Liver lysates from HFD-fed *Bach1*<sup>LKO</sup> and *Bach1*<sup>fl/fl</sup> mice were immunoprecipitated with anti-PTP1B antibodies and subjected to immunoblot analysis with antibodies against IR-β (upper). The quantification data of IR-β protein levels was shown (lower) (*n* = 6 biological replicates). **b** Representative immunostaining of IR-β-cherry (red), PTP1B-D181A-BFP (purple), and GFP (green) in HepG2 cells with or without BACH1-GFP overexpression after being treated with 100 nM insulin for 10 min. Nuclei were stained with DAPI (blue) (upper). Colocalization rates were calculated and showed (lower) (Scale bar = 20 μm, *n* = 3 biological replicates). **c** Left: Western blot analysis of the phosphorylation and total protein levels of IR-β, AKT, GSK-3β, and FOXO1 in the presence or absence of insulin (100 nM, 10 min) in primary hepatocytes infected with AdBACH1 or AdGFP and then transfected with PTP1B-siRNAs. Right: Phosphorylated protein levels were normalized to total protein (*n* = 3 biological replicates). Statistical analysis was performed by two-way ANOVA followed by Tukey post hoc tests for (**a**) and (**c**). Data are presented as mean values ± SD. Source data are provided as a Source Data file.

and pre-adipocyte 3T3-L1 (Cat# SCSP-5038)were purchased from the Chinese Academy of Sciences and cultured in DMEM supplemented with 10% FBS. Short tandem repeat (STR) profiling of HepG2, C2C12, and 3T3-L1 cells was tested. All cell lines were confirmed to be mycoplasma free with a mycoplasma detection kit and treated with Mycoplasma Elimination Reagent for the prevention of mycoplasma contamination.

All cells were cultured in a humidified atmosphere containing 5% $CO_2$ at 37 °C. Palmitic acid (PA) powder (Sigma–Aldrich, St. Louis, MO, USA, #P0500) was dissolved in 0.01 M NaOH to make a stock solution. The PA stock solution was diluted by mixing the indicated culture medium with 25% BSA (Sigma–Aldrich, St. Louis, MO, USA, #A9576) to make a PA solution. Oleic acid (OA) (Sigma–Aldrich, St. Louis, MO, USA, #O1008) was dissolved in 0.01 M NaOH to the indicated concentration. 1 mM PA or 1 mM OA for 12 h were used to induce lipid accumulation in primary hepatocytes. To stimulate insulin signaling, hepatocytes were stimulated with insulin 100 nM for the indicated time.

### Immunofluorescence
HepG2 cells were plated in 35 mm glass-bottom dishes at 90% confluence. The next day, cells were fixed with 4% PFA in PBS for 15 min and then washed three times with PBS. Cells were treated with 0.1% Triton X-100 in PBS for an additional 15 min at RT to penetrate the cell membrane. After that, cells were blocked with 10% donkey blocking serum before incubation with primary antibodies at 4 °C overnight: anti-LC3 (Cell Signaling Technology, #3868, diluted 1:1000), anti-ERp57 (Proteintech, Rosemont, IL, #15967-1-AP, dilution 1:200), anti-BACH1 (Proteintech, Rosemont, IL, #14018-1-AP, dilution 1:200), anti-GFP (Proteintech, Rosemont, IL, #50430-2-AP, dilution 1:200). Then, cells were washed and incubated with appropriate secondary antibodies in blocking reagent for 2 h at RT: 488 nm-conjugated anti-mouse secondary antibody, 594 nm-conjugated anti-rabbit secondary antibody, or 647nm-conjugated anti-rabbit secondary antibody. The isotype antibodies were used as the controls. After rinsing with PBS, nuclei were co-stained by DAPI in PBS for 15 min. Images were captured with a Leica DMI6000B microscope. Immunofluorescence quantification was performed by NIH Image J software[57]. Representative images were chosen to reflect the mean value of quantitative data. 4 - 5 independent experiments were calculated for each group.

### Constructions and transfections
The cDNA fragments that encoded full-length human *BACH1* or mutant versions of the sequence lacking portions of the BTB or Bzip domain with FLAG-tag were amplified with appropriate sets of primers and cloned into the pcDNA3.1 plasmid[25]. The cDNA fragments that encoded full-length mouse *Bach1* or mutant versions of the sequence lacking portions of the BTB or Bzip domain with HA-tag were amplified with appropriate sets of primers and cloned into the P-CMV plasmid[59]. The human PTP1B-HA and human IR-β-GFP plasmids were purchased from MiaoLing Plasmid Platform (Wuhan, China). The cDNA fragments that encoded D181A mutation human PTP1B (PTP1B-D181A) were cloned into the pLV3-CMV-MCS-TagBFP construct (MiaoLingBio, Wuhan, China, P50405). The cDNA fragments that encoded human IR-

β were cloned into the pCDNA3.1-Cherry construct (MiaoLingBio, Wuhan, China, P1737). HEK293T cells cultured in 6-well plates were transfected with 2 μg plasmids per well using jetPEI (Polyplus-transfection SA, Strasbourg, France, #24765-1) to determine protein–protein interactions.

For PTP1B knockdown, primary hepatocytes cultured in 6-well plates were transfected with 20 nmol/L mouse PTP1B-targeted small interfering RNAs (siRNA- PTP1B; designed and synthesized by Gene Pharma, Shanghai, China.) or with control random siRNA per well using Lipofectamine™ 3000 Transfection Reagent (Gibco, Carlsbad, CA, #L3000075) according to the manufacturer's instructions. The sequences of siRNA are listed in Supplemental Table S5.

To induce overexpression of *BACH1* or *BACH1* that lacked BTB domain in hepatocytes, we used a GFP-tagged recombinant adenovirus encoding full-length *BACH1* (AdBACH1; Hanbio Biotech, Shanghai, China) or a mutated version of human *BACH1* that lacked BTB domain (AdBACH1<sup>ΔBTB</sup>; Hanbio Biotech, Shanghai, China). To suppress *Bach1* expression in hepatocytes, we used a recombinant adenovirus encoding a *Bach1*-specific short hairpin RNA (sh*Bach1*; Hanbio Biotech, Shanghai, China), the sequences of shRNA are listed in Supplemental Table S5. A recombinant adenovirus expressing green fluorescent protein (AdGFP) or a null adenoviral vector (Control shRNA) was used as a control for *BACH1* overexpression and knockdown experiments, respectively. Hepatocytes were infected (MOI-20) with adenoviruses for 48 h. No evidence of cellular toxicity was detected.

### Immunoprecipitation
Liver tissues or cells were lysed in lysis buffer (20 mmol/L Tris-HCl pH 7.5, 150 mmol/L NaCl, 1% Triton X-100, 1 mmol/L EDTA, 1 mmol/L EGTA, 2.5 mmol/L sodium pyrophosphate, 1 mmol/L β-Glycerolphosphate, 50 mmol/L NaF, 1 mmol/L $Na_3VO_4$ and supplemented with protease inhibitor cocktail). Lysate was incubated with the primary antibody of the target protein at 4 °C overnight with rotation. The next day, lysates were incubated with Protein A/G PLUS -Agarose (Santa Cruz, Dallas, Texas, USA, #sc-2003) for 3 h at 4 °C. Then, lysates were washed with lysis buffer for three times before elution with 2 × SD sample buffer and boiled at 100 °C for 10 min, and immunoblotting was performed via standard protocol[24].

### Glutathione s-transferase (GST) pull-down assays
The direct interaction between BACH1-PTP1B and BACH1-IR-β was determined using GST precipitation assays[59]. In brief, the HA-tagged PTP1B or GFP-tagged IR-β plasmids (10 μg per dish) were transfected into 293 T cells using jetPEI (Polyplus-transfection SA, Strasbourg, France, #24765-1). The PTP1B and IR-β proteins were purified 48 h after transfection. The GST-tagged BACH1 was subcloned into the Escherichia coli expressing pGEX vector and expressed in the Escherichia coli strain BL21. Bacterial cultures were grown at 37 °C, induced with 0.1 mM isopropyl-b-Dthiogalactopyranoside (IPTG), and then harvested by centrifugation at 12,000 g for 10 min at 4 °C. The GST-tagged BACH1 was expressed and purified with glutathione-Sepharose 4B beads. Purified GST-tagged recombinant proteins BACH1 and purified HA-tagged PTP1B or GFP-tagged IR-β proteins were then incubated together for 2 h at 4°C, followed by three times wash. Conjugated

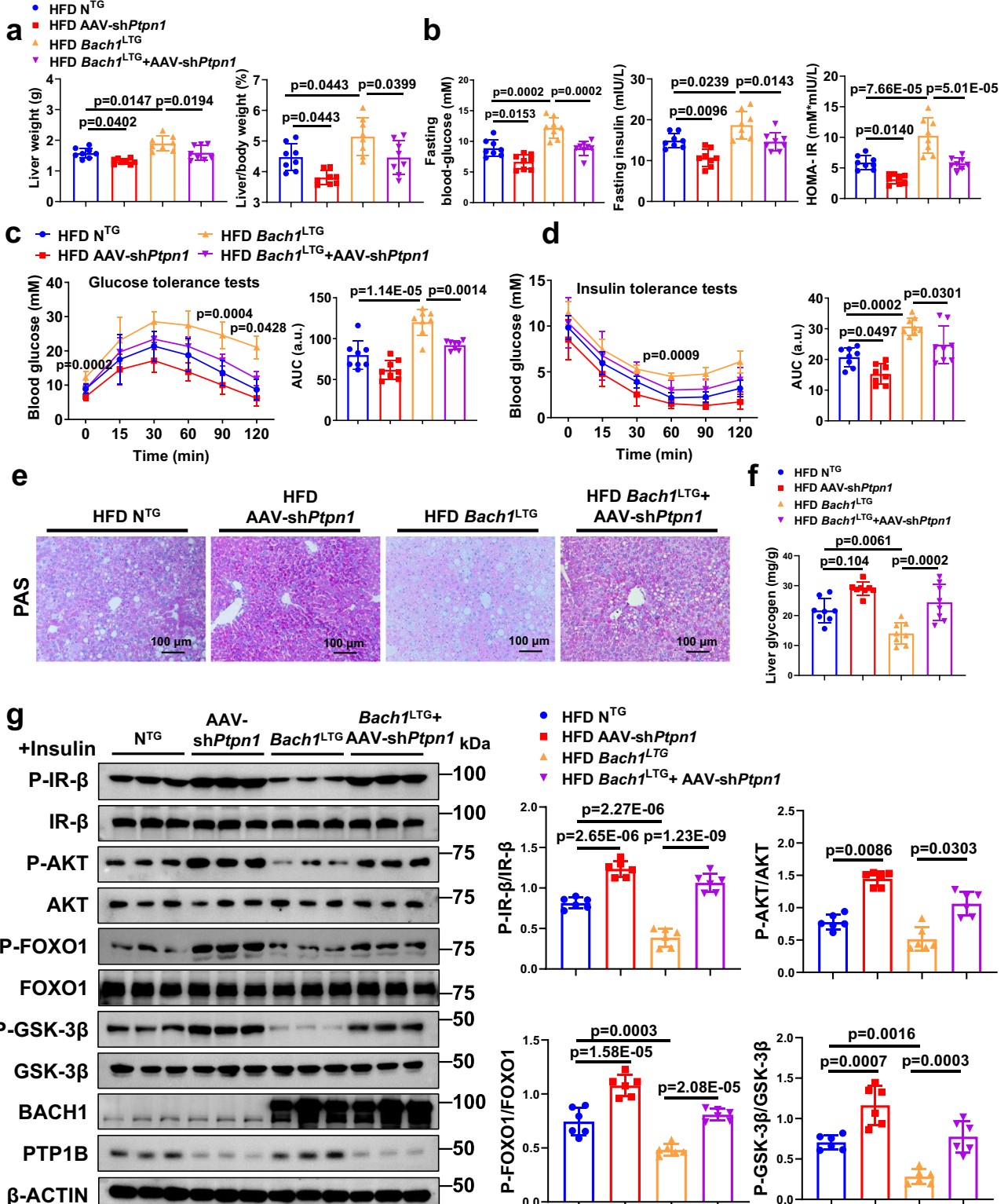

**Fig. 7 | BACH1 aggravates insulin resistance through a PTP1B-dependent manner. a** Liver weights (left) and liver/body weight ratio (right) of male N[TG] and *Bach1*[LTG] mice injected with AAV-TBG-GFP or AAV-sh*Ptpn1* and subjected to HFD challenge for 12 weeks (*n* = 8 mice per group). **b** Fasting blood glucose levels (left), fasting insulin levels (middle), and corresponding HOMA-IR index (right) of the above 4 groups of mice (*n* = 8 mice per group). **c** Left: GTTs were measured at week 12 of HFD feeding (*n* = 8 mice per group). Right: The AUC was used to quantify the GTT results. **d** ITTs were measured at week 12 of HFD feeding (*n* = 8 mice per group). Right: the AUC for ITTs. **e, f** PAS staining (**e**) and glycogen assay (**f**) were carried out to determine the glycogen content in the mouse livers. Representative

PAS staining was shown in (**e**) (*n* = 8 mice per group, Scale bar = 100 μm). **g** Left: Western blot analysis of essential markers of the insulin pathway (IR-β, AKT, FOXO1, and GSK-3β) in the livers from the above 4 groups of mice after insulin administration. Right: Phosphorylated protein levels were normalized to total protein (*n* = 6 mice per group). Statistical analysis was performed by two-way ANOVA followed by Tukey post hoc tests for **a**, **b**, 0 min, and 90 min in (**c**), (**d**), (**f**), and (**g**), and by two-way ANOVA followed by Kruskal–Wallis test with Dunn multiple comparisons test for 120 min in (**c**). Data are presented as mean values ± SD. Source data are provided as a Source Data file.

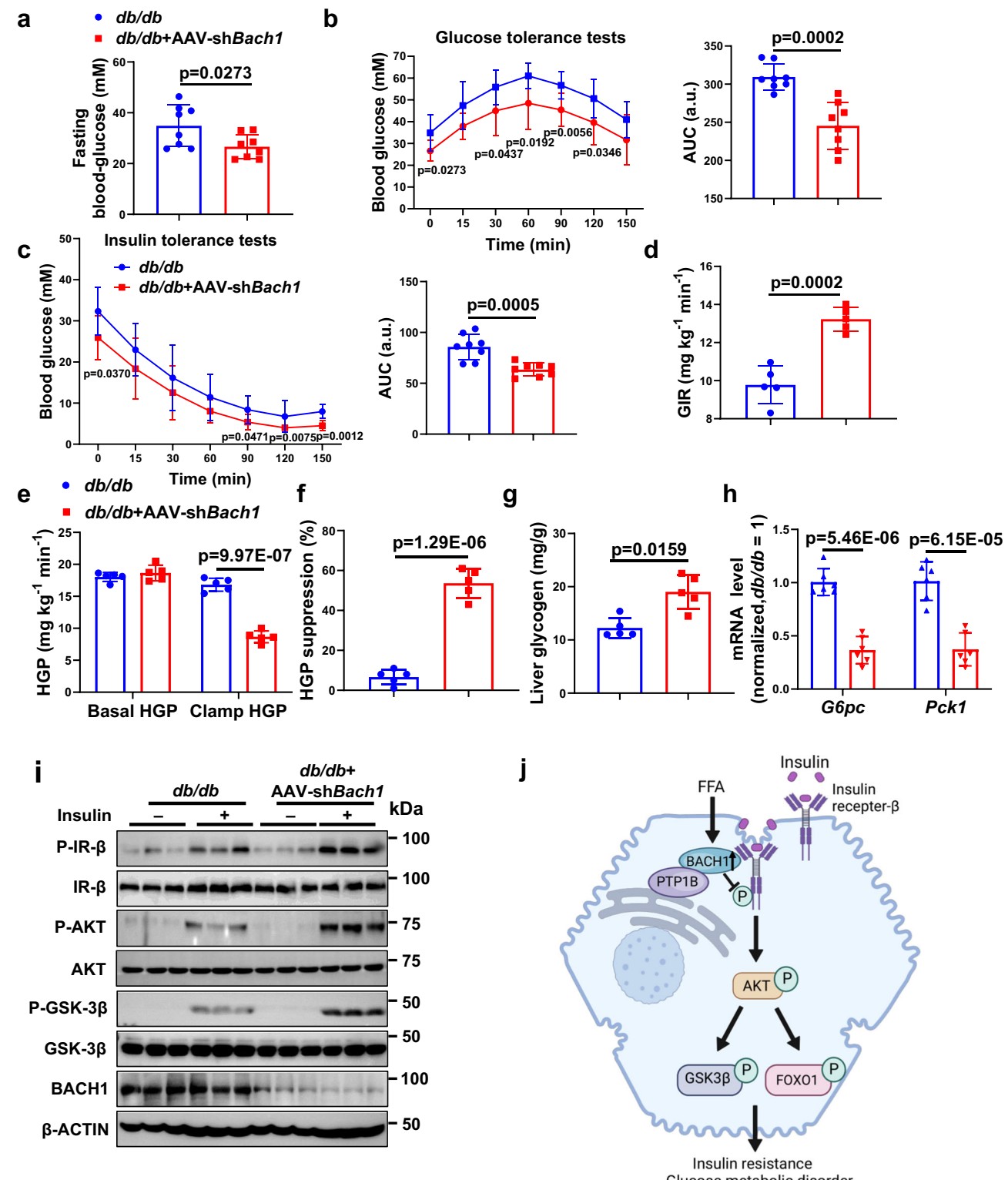

proteins were then eluted by SDS loading buffer, and examined by western blot analysis.

## Western blotting analysis

Murine liver tissue or cells were homogenized in RIPA buffer supplemented with PMSF (Beyotime, Shanghai, China, #ST505) and protease inhibitor cocktail (MedChemExpress, Shanghai, China #HY-K0010) using a bullet blender (Jingxin, Shanghai, China, #Tissuelyser-24). After centrifuged for 15 min at 12000 × $g$ at 4 °C, equal amounts of protein were loaded onto an 8%–12% SDS-PAGE gel for separation and transferred to a PVDF membrane (Millipore, Billerica, MA, #IPVH00010). Membranes were blocked in a solvent containing PBS/0.1% (v/v) Tween-20 (PBS-T) and probed with primary antibodies in a primary antibody dilution buffer (Beyotime, Shanghai, China, # P0256). They were then incubated with horseradish peroxidase-conjugated secondary antibody in PBS-T with 5% (w/v) skimmed milk (Sangon Biotech, Shanghai, China, #A600669) and developed using ECL detection reagent (Tanon, Shanghai, China, #180-5001)[24]. Antibodies against

**Fig. 8 | Knockdown of BACH1 ameliorates hyperglycemia and insulin resistance in *db/db* diabetic mice. a** Male *db/db* mice were injected with AAV control or AAV-sh*Bach1* for four weeks, and the serum fasting blood glucose levels of *db/db* mice were examined (*n* = 8 mice per group). **b** GTTs were measured (*n* = 8 mice per group) (left). The AUC was used to quantify the GTT results (right). **c** ITTs were measured (*n* = 8 mice per group) (left). The AUC was used to quantify the ITT results (right). **d** The glucose infusion rate (GIR) was calculated for the last 40 min of insulin infusion (*n* = 5 mice per group). **e** Basal and clamped rates of hepatic glucose production (HGP) were calculated (*n* = 5 mice per group). **f** Suppression of HGP in *db/db* mice receiving either AAV control or AAV-sh*Bach1* was determined (*n* = 5 mice per group). **g** A glycogen assay was carried out to determine the glycogen content in the livers of *db/db* mice receiving either AAV control or AAV-sh *Bach1* (*n* = 5 mice per group). **h** mRNA expression of *Pck1* and *G6pc* was measured in the

livers of *db/db* mice receiving either AAV control or AAV-sh*Bach1* by qRT-PCR (*n* = 6 per group). **i** Western blot analysis of essential markers of insulin signaling (IR-β, AKT, and GSK-3β) in the liver tissues from *db/db* mice receiving either AAV control or AAV-sh*Bach1* after injected i.p. with insulin after overnight fasting (*n* = 3 per group). **j** Working model of the role of hepatic BACH1 in insulin signaling regulation. The increased BACH1 expression induced by excessive nutrient intake impaired insulin signaling and aggravated insulin resistance by facilitating the binding of PTP1B and IR-β. Statistical analysis was performed by two-tailed Student's *t*-test for **a**, **b**, 0 min, and 90 min in (**c**), (**d**–**f**), and (**h**), by two-tailed Mann–Whitney *U* test for 120 min and 150 min in (**c**), and for (**g**). Data are presented as mean values ± SD. Source data are provided as a Source Data file. Figure 8j was created with Biorender under a paid subscription (agreement number: IZ263QDPCW).

---

target proteins were used for immunoblotting and the relevant antibody information is listed in Table S6. Uncropped and unprocessed scans of the most important blots are supplied in the Source Data file.

### BODIPY 493/503 staining

HepG2 cells were pretreated by 1 mM OA for 12 h to induce lipid accumulation and then incubated for 30 min in a 60% working solution with 1 mg/mL BODIPY (MedChemExpress, Shanghai, China). After washing with PBS, images were captured with a confocal microscope (Leica DMI6000B microscope).

### Statistical analysis

All values are expressed as the mean ± SD. The distribution of the data was assessed using the Shapiro–Wilks normality test. For two-group comparisons of normally distributed data, we applied a two-tailed Student's *t*-test for data of similar variances, or with Welch's correction if equal standard deviations are not assumed through an *F* test. For more than two-group comparisons, the Brown-Forsythe test was used to assess similar variances, followed by ordinary one- or two-way ANOVA followed by Tukey post hoc tests, respectively. Biological experimental replicates in each group are shown in the figure legends, and the exact *p*-values of the results are specified in the figure. No animals were excluded from the current study. $p < 0.05$ was considered statistically significant. All statistical analyses were performed by GraphPad Prism 8.0 software (GraphPad Software, San Diego, CA, USA) and Image J 1.50b software (National Institutes of Health, Montgomery County, Maryland, United States).

### Reporting summary

Further information on research design is available in the Nature Portfolio Reporting Summary linked to this article.

## Data availability

There are no restrictions on data availability. All data supporting the findings of this study are available within the main text, supplementary information, and Source data. The RNAseq data of *BACH1* mRNA expression in the liver tissue of lean individuals or individuals with obesity is under the accession number GSE192742. The RNAseq data of BACH1 mRNA expression in the liver tissue of mice following a high-sucrose-and-high-fat diet (HSD) is under the accession number GSE182365. If needed, contact JY.J. (jinjiayu@fudan.edu.cn) for the original data described in the paper. Contact D.M. (dmeng@fudan.edu.cn) for requesting *Bach1*fl/fl and *Bach1*TG mouse strain, and all other plasmids or reagents described in this article. Source data are provided with this paper.

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

## Acknowledgements

This work was supported by the Great Program (92068202 to D.M. and 82220108020 to D.M.) of the National Natural Science Foundation of China, Program of Shanghai Academic/Technology Research Leader (20XD1400600 to D.M.), and the Natural Science Foundation of Shanghai (22ZR1415100 to XL.Z.). We gratefully

acknowledge Prof. Mingliang Zhang (Shanghai Jiao Tong University Affiliated Sixth People's Hospital, China) for providing guidance on hyperinsulinemic-euglycemic clamping.

## Author contributions

D.M. and XL.Z. conceived and designed the project. JY.J. performed most experiments and result analyses. YQ.H. assisted and repeated many some of the experiments. YQ.H. and JY.G. performed some animal experiments and assisted result analyses. C.X. provided liver tissues of patients. Q.P. helped with some bioinformatics analysis. ZY.Q. performed some WB and qRT-PCR experiments. XX.W., QH.L., SY.M., JY.L., N.J., JH.M., XH.W., LD.J., and E.O. provided valuable comments. D.M. and JY.J. wrote the manuscript. D.M. supervised the whole study.

## Competing interests

The authors declare no competing interests.
