## [Peer Review File · Nature Communications]

BACH1 Controls Hepatic Insulin Signaling and Glucose HomeostasisREVIEWER COMMENTS

Reviewer #1 (Remarks to the Author):

This manuscript appears interesting and important to the fields of metabolism, with a completely new mechanism of insulin signal transduction regulation by transcription factor BACH1. Very surprisingly, the regulation did not involve transcription regulation.

1. It will be very important to clearly establish that transcription regulation is not involved in what authors observed. Toward this end, authors can compare insulin sensitivity and IR/PTP1B binding by expressing wt BACH1, BACH1 lacking BTB, and BACH1 lacking bZip in BACH1 KO cells or the liver. Also it will be important to show that BTB alone can execute the regulation authors suggested. To exclude transcriptional regulation, HO-1 expression can be also compared because it is highly sensitive to the transcription factor activity of BACH1.
2. BACH1 protein increase in the model liver and human patients seems very interesting. However, the assays are not so quantitative. To further validate, RT-PCR is needed. Also, for immunohistochemical analysis, negative controls are essential, such as control primary antibodies and staining of knockout tissues. Also for human patients, western blot analysis allows more quantitative evaluation.
3. If the protein interactions are valid, a fraction of BACH1 protein is expected to be bound to cell membrane, which was not clear from the results provided. Also, insulin treatment is expected to alter BACH1 localization. For example, how BACH1 is altered in Fig. 6. In sup fig 5, BACH1 was in the cytoplasmic region. Such a distribution altered by BTB deletion? By the way, cytoplasmic/nuclear dynamics of BACH1 was reported by others, which should be cited properly (pubmed 15175654, 14504288).
4. Related to the issue 1, the interaction mediated by BTB domain needs to be elaborated further. Is BTB domain sufficient or not, requiring other regions of BACH1? Which protein does BTB domain directly bind?
5. Fig. 1A is difficult to interpret. First, these SNPs can affect nearby genes. Second, it is not shown how significant these SNPs are when viewed genome-wide. Third, authors explain that 10kb regions up and down of BACH1 gene were analyzed. However, BACH1 gene itself spans more than 150 kb DNA. Careful analysis and better, correct figure are needed here.
6. A band can be detected in BACH1 KO samples. Is this a non-specific band?
7. The regulation of insulin sensitivity in myoblasts and adipocytes are important and results should be shown.
8. Some discussion may be helpful for readers. Does BACH1 protein increase in other disease models?

Reviewer #2 (Remarks to the Author):

This manuscript revealed the role of BACH1 in insulin sensitivity, and the result demonstrated a novel function for hepatic BACH1 in the regulation of insulin sensitivity and glucose homeostasis, providing a potential target for the treatment of metabolic disorders, the advices were as follow :
Major :

- 1.The insulin resistance should be detected by Hyperinsulinemic Euglycemic Clamp;
- 2.In this manuscript, the in vitro experiment revealed BACH1 interacted with PTP1B and IR- β and its mechanism by immunoprecipitation, GST-pull down and immunofluorescence, but there maybe some difference between in vitro and in vivo experiment. We suggested they should investigate BACH1 interacted with PTP1B and IR- β by Small living animal imaging technology in animal models;
- 3.Lipophagy is closely related to insulin resistance, FOXO1 and GSK-3 β . We suggested that some experiments about the effect on lipophagy in BACH1 and its related signal pathways should be supplemented, and also in discussion.

Minor :

The language and grammar style should be polished by native English speaker.

conclusion : Major revision

Reviewer #3 (Remarks to the Author):

The authors report that BACH1 dampens insulin signaling in the liver, acting beyond its traditional role as a transcription factor. They show that BACH1 expression is increased in several models of obesity in humans and mice. In cells they demonstrate that treatment of hepatocytes with fatty acids induces BACH1 expression, providing a potential link between BACH1 and insulin resistance in an obesogenic environment. This interesting and thorough study sheds light on a novel role of BACH1 in insulin signaling, which may represent a potential mechanism of hepatic insulin resistance.

Major comments:

1) The authors perform a GWAS identifying 94 genetic variants within 5 kb of Bach1 associated with obesity or metabolic disorders. The authors should justify their selection of 5 kb, as this is a very large window and therefore may be less specific to BACH1 effects than say 2 kb. Furthermore, the authors include obesity with other metabolic disorders. Is BACH1 associated with insulin resistance independently of obesity per se? What are the results if obesity is not included as a phenotype? This would give a better idea of the causal effects of BACH1 on metabolic disease, since the authors predict that BACH1 changes in response to obesity.

2) The authors use HOMA-IR, ITT and GTT as surrogate measures of whole-body insulin sensitivity, and liver glycogen content and gene expression of gluconeogenesis enzymes as liver-specific measures. Unfortunately, direct measures of hepatic insulin sensitivity were not obtained, e.g. rate of hepatic glucose production and/or glycogen synthesis during a hyperinsulinemic euglycemic clamp or in isolated hepatocytes. The authors should justify the use of these indirect measures for liver insulin sensitivity, and tone down the conclusions about a definitive role of BACH1 in hepatic insulin resistance. This should be included in the discussion as a limitation to the study. On a related note, line 248 says BACH1 overexpressing livers have lower glycogen storage "capacity". However, glycogen storage capacity was not actually measured, only glycogen content. This should be corrected for accuracy.

3) The authors show that BACH1 blunts insulin signaling in hepatocytes through interaction of its BTB domain with insulin receptor β and the phosphatase PTP1B. However, these experiments were performed at only one (maximal) dose of insulin (100 nM). Because submaximal insulin doses were not used as well, from a pharmacological perspective it is not definitive whether insulin sensitivity or max response is shifted. This should be specified in the manuscript.

4) The authors show that BACH1 modifies insulin signaling, and that modification of BACH1 expression affects surrogate measures of insulin sensitivity, at least partly dependent on PTP1B as shown by PTP1B knockdown. While these experiments are elegant and informative, I do not believe that they definitively link changes in hepatocyte insulin signaling with changes in insulin sensitivity in the AAV8-TBG mouse models. For example, what are coincidental changes in liver protein expression as a result of the actions of BACH1 as a transcription factor? Does deletion of the BTB domain affect its activity as a transcription factor? This should be discussed.

5) The authors overexpress BACH1 in liver by tail vein injection of AAV8-TBG-Cre into BACH1 fl/fl mice. To delete BACH1, AAV8-Bach1-shRNA is injected. AAV8-TBG is known to have good hepatocyte specificity. Indeed the authors show total protein levels of BACH1 are unaffected in several other tissues, only in liver. AAV8-TBG-GFP is used as a negative control, which is appropriate since AAV8 is known to elicit an immune response in hepatocytes. The authors should

perform IHC of BACH1 in liver sections to demonstrate Cre efficiency (% of cells and location, e.g. is the periportal region more affected and periphery less affected).

6) For the glucose tolerance test, the authors perform an I.P. GTT rather than oral gavage. They should list this as a limitation, since I.P. GTT causes a poor insulin secretory response compared to oral gavage, and therefore may be less relevant as a measure of insulin sensitivity. Furthermore, the tolerance tests are dosed by body weight rather than lean mass, which may confound interpretation as obese animals will have a higher delivery of glucose or insulin than lean animals.

7) The mouse experiments were performed in males only. The authors should justify their choice of males and exclusion of females. This should be listed in the discussion as a study limitation.

Minor comments:

- 1) Fig 6B magnification too small, inset magnification too small
- 2) Fig 7G BACH1 blot overexposed, cropped too tightly
- 3) Fig 8 working model too much interpretation in figure legend—"These findings suggest that BACH1 could be an attractive therapeutic target for the treatment of diabetes, potentially facilitating improved insulin sensitivity and glucose tolerance."
- 4) Line 394—Bach1 should be capitalized
- 5) Line 429—should read "of BACH1"
- 6) Line 849—should be "In brief" or "Briefly"

Reviewer #4 (Remarks to the Author):

This is a very interesting paper showing original results of the role of BACH1 protein in insulin signalling and glucose homeostasis in the liver of mice. The authors used different strategies (knockout, gain-of-function, cell experiments) to demonstrate that BACH1, through its interaction with PTP1B and the insulin receptor beta, can inhibit insulin stimulation in hepatocytes and reduce glucose tolerance, especially under a high fat diet condition. Although their results are convincing about the important role that may have BACH1 in glucose metabolism in mice, I have concern about supporting evidence in humans.

1. First of all, there is no significant SNPs at the traditional GWAS p-value threshold in the BACH1 gene region. The threshold that should be used is $P < 5 \times 10^{-8}$. All SNPs in Fig.1 are below that threshold. Are there other relevant results in the GWAS Catalog for that gene?
2. Again regarding human results, I am not sure we can say that the expression of BACH1 in the liver of metabolically healthy and unhealthy are significantly different since an outlier seems to bias the results.
3. Since the results shown in human are not convincing, I would try to add more human results published in the literature regarding the potential role of BACH1 in metabolic disorders, if there is more than what was said in the paper (introduction, discussion).
4. Minor comment: Maybe use 'BACH1 Regulates Insulin Signalling' rather than 'Resistance' in the running title.

Response to Reviewers

Manuscript No. NCOMMS-23-02730-T R1

**“BACH1 Controls Hepatic Insulin Signaling and Glucose Homeostasis”
by Dr. Jiayu Jin et al.**

Reviewer #1

Comments for the Authors

This manuscript appears interesting and important to the fields of metabolism, with a completely new mechanism of insulin signal transduction regulation by transcription factor BACH1. Very surprisingly, the regulation did not involve transcription regulation.

Issues needing clarification:

1. It will be very important to clearly establish that transcription regulation is not involved in what authors observed. Toward this end, authors can compare insulin sensitivity and IR/PTP1B binding by expressing wt BACH1, BACH1 lacking BTB, and BACH1 lacking bZip in BACH1 KO cells or the liver. Also, it will be important to show that BTB alone can execute the regulation authors suggested. To exclude transcriptional regulation, HO-1 expression can be also compared because it is highly sensitive to the transcription factor activity of BACH1.

Response: Thank you for your important suggestions. We have performed WB and IP experiments to compare insulin sensitivity and IR- β /PTP1B binding by expressing WT BACH1, BACH1 lacking BTB, and BACH1 lacking Bzip in BACH1 KO hepatocytes from *Bach1*^{LKO} mice. We found that BACH1 deficiency led to elevated phosphorylation levels of IR- β , AKT, GSK-3 β , and FOXO1 in primary hepatocytes isolated from *Bach1*^{LKO} mice upon insulin stimulation, which were significantly inhibited by vectors coding for HA-tagged full BACH1 sequence (BACH1-HA) and for truncated sequences lacking the Bzip domain (BACH1 ^{Δ Bzip}-HA), but not by vectors coding for truncated sequences lacking the N-terminal BTB domain of BACH1 (BACH1 ^{Δ BTB}-HA) (new Fig. 5A). These results suggested that the BTB domain, but not the Bzip domain of BACH1 is essential for repressing the insulin signaling by BACH1. Because HO-1 expression is highly sensitive to the transcription factor activity of BACH1, we also determined the HO-1 expression in the above experiment. We found that HO-1 expression in BACH1 KO cells was significantly inhibited by full BACH1, but not by BACH1 ^{Δ Bzip} or BACH1 ^{Δ BTB}, which suggests that both the BTB domain and the Bzip domain of BACH1 are important for the regulation of HO-1. This indicates that deletion of the BTB domain of BACH1 also affects its activity as a transcription factor. It was reported that hepatocyte conditional HO-1 deletion in mice evoked resistance to diet-induced insulin resistance and reduced diet-induced fatty liver disease ¹. *Bach1* ablation exerts protective effects against steatohepatitis presumably via HO-1 induction ². Thus, the transcriptional regulatory effect of BACH1 may also be involved in regulating insulin signaling, and we cannot rule out this possibility. Similarly, we found that the IR- β /PTP1B binding was significantly decreased in hepatocytes from *Bach1*^{LKO} mice compared with that in cells from the control mice by co-immunoprecipitation (co-IP) assay, while the decreased binding in BACH1 KO hepatocytes was significantly enhanced by the full length of

BACH1 and BACH1^{ΔBzip} but not by BACH1^{ΔBTB} (Response Fig.1A), indicating that BTB domain of BACH1 is critical for the binding of IR-β/PTP1B promoted by BACH1.

Moreover, we examined the role of the BTB domain of BACH1 alone in the regulation of insulin signaling and IR-β/PTP1B binding. In contrast to what we previously suggested, BACH1 BTB domain alone cannot inhibit insulin signaling nor bind to PTP1B or IR-β, according to the new results obtained with WB and IP (Response Fig. 1B). Previous study found BACH2 required both BTB and the region between BTB and bZip for the focus formation upon oxidative stress³. It is possible that the BTB domain of BACH1 alone is unstructured and require other regions (the region between BTB and bZip) of BACH1 to bind to PTP1B or IR-β. Taken together, these results indicate that the transcription regulation might also be involved in the function of BACH1 in regulating insulin signaling. We have added it to the Results and Discussion section (lines 227-247 on page 11-12, lines 388-394 on page 18, marked in blue) in the revised manuscript.

The new figures are displayed in new Fig. 5A and Response Figure 1 as follows:

Fig.5

Fig. 5. BACH1 interacts with PTP1B and IR-β by the BTB domain.

(A) Left: Western blot analysis of the phosphorylation and total protein levels of IR-β, AKT, and GSK-3β in primary hepatocytes isolated from *Bach1*^{LKO} mice. Hepatocytes were transfected with vectors coding for HA-tagged versions of the full *Bach1* sequence (BACH1-HA) or mutant sequences lacking the Bzip domain (BACH1^{ΔBzip}-HA) or BTB domain (BACH1^{ΔBTB}-HA) with or without insulin (100 nM) stimulation for 10 min. Right: Phosphorylated protein levels were normalized to total protein.

Response Figure 1.

(A) The Co-IP analyses in primary hepatocytes isolated from *Bach1*^{LKO} mice. Left: Hepatocytes were transfected with vectors coding for BACH1-HA or BACH1^{ΔBzip}-HA or BACH1^{ΔBTB}-HA. Cell lysates were immunoprecipitated with anti-IR-β antibody and subjected to immunoblot analysis with antibodies against PTP1B. Right: PTP1B protein levels were quantified normalized to total protein (data are presented as the mean±SD). Statistical analysis was performed by one-way ANOVA followed by Tukey post hoc tests. **(B)** The Co-IP analyses of the interaction of BTB-HA/PTP1B and BTB-HA/IR-β. Mice primary hepatocytes were transfected with BTB-HA. Hepatocytes with or without treatment with insulin (100 nM, 10 mins), were lysated and immunoprecipitated with HA. Immunoblotting was used to detect IR-β and PTP1B. **(C)** Western blot analysis of the phosphorylation and total protein levels of IR-β, AKT, and GSK-3β in primary hepatocytes transfected with BACH1-HA or BTB-HA a with or without treatment with insulin (100 nM, 10 mins).

2.BACH1 protein increase in the model liver and human patients seems very interesting. However, the assays are not so quantitative. To further validate, RT-PCR is needed. Also, for immunohistochemical analysis, negative controls are essential, such as control primary antibodies and staining of knockout tissues. Also for human patients, western blot analysis allows more quantitative evaluation.

Response: Thank you for your valuable suggestions. As suggested, we have performed qRT-PCR to detect BACH1 mRNA expression in the livers of HFD mice and NAFLD patients, as well as Western blot analysis of hepatic BACH1 in patients. As shown in new Fig. 1B-C and 1G, BACH1 protein and mRNA expression are significantly upregulated in the livers of NAFLD patients as well as HFD mice. We also provided negative controls for immunohistochemical staining of BACH1. The sections of the livers of NAFLD patients incubated with rabbit IgG (10 μg/mL) were used as a negative control for BACH1 staining (new Fig. S1B). These data indicate the specificity

of the BACH1 antibody and the efficiency of *Bach1* knockout in mice hepatocytes. The new figures are displayed in new Fig. 1 B-C and 1G, Fig. S1B, and Fig. S2C as follows:

Fig. 1

Fig. 1. BACH1 is elevated in the livers of patients with NAFLD and obese mice.

(A) *BACH1* mRNA expression in the livers of lean or obese individuals in the public RNA-sequencing database (GEO accession no. GSE192742). (B-C) *BACH1* mRNA expression (B) and protein expression (C) in the liver tissues from healthy subjects (Normal) and NAFLD patients (n=6). (D) The representative images of H&E, PAS staining, and immunohistochemistry assay of BACH1 from liver tissues of healthy subjects and NAFLD patients. Scale bars=100 μ m. (E) Quantitative data of

immunohistochemistry assay of BACH1 in the liver tissues (n=6). **(F)** *Bach1* mRNA expression in liver cells from mice following a high-sugar diet (HSD) compared with chow diet (CD) (GEO accession no. GSE182365). **(G-H)** *Bach1* mRNA **(G)** (n=8) and protein expression **(H)** (n=5) in the liver tissues from male CD- and HFD-fed mice. **(I-J)** BACH1 protein expression in male *ob/ob* mice **(I)** (n=4) and *db/db* mice **(J)** (n=3). **(K)** Primary hepatocytes were treated with oleic acid (OA, 1 mM) for 12 hours and then subjected to immunoblot analyses to determine the BACH1 protein expression (n=3). Statistical analysis was performed by Mann-Whitney U test for A, E, G, and by Student's t test for B, F.

Fig. S1. BACH1 is elevated in the livers of patients with NAFLD and obese mice. **(B)** Immunohistochemical staining of BACH1 in livers of NAFLD patient. Rabbit IgG (10 μg/mL) was used as a negative control.

Fig. S2. BACH1 is specifically knocked down in the mouse livers. **(C)** Left: The representative images of immunohistochemistry assay of BACH1 in *Bach1^{fl/fl}* and *Bach1^{LKO}* mouse liver samples. Right: Quantitative data of immunohistochemistry assay of BACH1 in the liver tissues (n=3).

3. If the protein interactions are valid, a fraction of BACH1 protein is expected to be bound to cell membrane, which was not clear from the results provided. Also, insulin treatment is expected to alter BACH1 localization. For example, how BACH1 is altered in Fig. 6. In sup fig 5, BACH1 was in the cytoplasmic region. Such a distribution altered by BTB deletion? By the way, cytoplasmic/nuclear dynamics of BACH1 was reported by others, which should be cited properly (pubmed 15175654, 14504288).

Response: We thank the Reviewer for this important question. As suggested, we have determined the BACH1 localization at various time points of insulin stimulation by immunofluorescent staining and Western blot analysis for membrane and cytoplasmic cell fractions in human HepG2 cells. Our results showed that localization of BACH1 was not altered at 1, 5, 15 or 30 min after insulin stimulation as evidenced by

immunofluorescent staining (new Fig. S10C). Moreover, BACH1 was not present on the cell membrane after insulin stimulation in the membrane fractions of cells by Western blot analysis (new Fig. S10D). These data suggest that BACH1 did not bind to the cell membrane and insulin treatment did not alter BACH1 localization. Finally, we found that the deletion of the BTB domain of BACH1 resulted in its nuclear accumulation (new Fig. S10B), which is consistent with a previous study⁴. Because BACH1 lacking BTB domain is rarely localized in the cytoplasm, these results also explain why BACH1 lacking BTB domain no longer bound to IR- β or PTP1B; this further support the notion that BTB domain of BACH1 is critical for the binding of IR- β /PTP1B.

It is known that upon insulin binding, the insulin receptor is activated by autophosphorylation. The activated insulin receptor is internalized and dephosphorylated by PTP1B, which is located on the cytosolic side of the endoplasmic reticulum⁵. Because the activities of PTP1B have very high turn-over rates, a substrate-trapping mutant of PTP1B (PTP1B-D181A) was used to detect insulin-induced interaction between PTP1B and IR- β ⁵. PTP1B-D181A retains the ability to bind to IR- β but cannot dephosphorylate its substrates. Localization of PTP1B-D181A and the IR- β in transfected HEK 293 cells was shown in a region of the cell surrounding the nucleus⁵, and this distribution is consistent with the association of PTP1B with the endoplasmic reticulum⁶. By using immunofluorescence assay, we found that BACH1 was detected in the cytoplasm, and enriched in the endoplasmic reticulum of HepG2 cells (new Fig. S10A). We then detected the localization of BACH1-GFP, PTP1B-D181A-BFP and IR- β -cherry in transfected HepG2 cells. Immunofluorescence staining showed co-localization of the three proteins primarily to the cellular space surrounding the nucleus (new Fig. 5C), which is consistent with the previous report about the distribution for PTP1B-D181A and IR- β ⁵. Upon insulin stimulation, overexpression of BACH1 resulted in enhanced interaction of PTP1B-D181A and IR- β in the perinuclear compartment (new Fig. 6B). Consistent with these results, loss of BACH1 reduced the interaction between IR- β and PTP1B upon insulin stimulation in the liver tissues of HFD-fed *Bach1*^{LKO} mice (Fig. 6A). Thus, we speculate that BACH1, PTP1B and IR- β co-localized in the endoplasmic reticulum in the basal state, overexpressed BACH1 facilitated PTP1B binding to internalized IR- β in the endoplasmic reticulum, and subsequently inactivated IR- β by PTP1B binding upon insulin stimulation. Indeed, we observed that inhibition of PTP1B attenuated BACH1-mediated suppression of insulin signaling both *in vivo* and *in vitro*, suggesting that BACH1 modulates insulin signaling, at least in part, via PTP1B.

We thank the Reviewer for bring our attention to these studies (pubmed 15175654, 14504288), and we have added them to the Introduction section (page 4, line 76-78; page 12, line 258-259) and the reference list (reference 22-23) in the Introduction of the revised manuscript (reproduced below):

“BACH1 nuclear export was triggered by heme and cadmium, and both heme- and cadmium-induced BACH1 nuclear export signals were dependent on chromosome region maintenance 1 (Crm1)^{4,7}.”

“The deletion of the BTB domain of BACH1 resulted in its nuclear accumulation (Supplementary Fig. 10B), which is consistent with a previous study⁴”

The new figures are displayed in new Fig. 6A-6B, Fig. S10A-10D as follows:

Fig.6

Fig. 6. BACH1 enhances the interaction between PTP1B and IR-β in response to insulin.

(A) The co-IP analyses of the interaction of IR-β and PTP1B in the liver lysates from HFD-fed male *Bach1^{LKO}* and *Bach1^{fl/fl}* mice. Mice were injected i.p. with insulin (100 nM) after overnight fasting. Liver lysates from HFD-fed *Bach1^{LKO}* and *Bach1^{fl/fl}* mice were immunoprecipitated with anti-PTP1B antibody and subjected to immunoblot analysis with antibodies against IR-β (upper). The quantification data of IR-β protein levels was shown (lower, n=6). **(B)** Representative immunostaining of IR-β-cherry (red), PTP1B-D181A-BFP (purple), and GFP (green) in HepG2 cells with or without BACH1-GFP overexpression after treated with 100 nm insulin for 10 min. Nuclei were stained with DAPI (blue) (upper). Colocalization rate were calculated and showed (lower). Scale bars=20 μm.

Supplementary Fig 10

Fig. S10. BACH1 ER localization is not affected by insulin stimulation but by BTB domain.

(A) Representative immunostaining of BACH1 (green) and ERp57 (red) in HepG2 cells. HepG2 cells were fixed and stained with anti-BACH1 and anti- ERp57 antibody. **(B)** HepG2 cells were transfected with BACH1-FLAG or BACH1^{ΔBTB}-FLAG for 48h and then fixed and stained with anti-FLAG antibody (green). **(C)** Representative immunostaining of BACH1 (green) in HepG2 cells upon insulin stimulation. HepG2 cells were treated with 100nM insulin for 0 min, 1 min, 5 min, 15 min and 30 min respectively and then fixed and stained with anti-BACH1 antibody. Nuclei of A-C were stained with DAPI (blue). Scale bars=10 μ m. **(D)** Western blot analysis of BACH1 in cytoplasm and plasma membrane of mouse primary hepatocytes. Cells were treated with 100nM insulin for 0 min, 1 min, 5 min, 10 min, and then the plasma membrane (PM) and cytosol (Cyt) were fractionated and analyzed by immunoblotting. The Na/K ATPase and HSP90 were used as PM and Cyt loading controls, respectively.

4. Related to the issue 1, the interaction mediated by BTB domain needs to be elaborated further. Is BTB domain sufficient or not, requiring other regions of BACH1? Which protein does BTB domain directly bind?

Response: Thank you for your important suggestions. We have performed Western

blot and coimmunoprecipitation (Co-IP) assay to determine whether insulin signaling and IR- β /PTP1B binding could be regulated by BTB domain of BACH1 alone in mouse primary hepatocytes. As shown in above Response Figure 1, BACH1 BTB domain alone cannot execute the regulation as we suggested, it can neither inhibit insulin signaling nor bind to PTP1B or IR- β . Previous study found BACH2 requires both BTB and the region between BTB and bZip for the focus formation upon oxidative stress³. It is possible that the BTB domain of BACH1 alone is unstructured and requires other regions (the region between BTB and bZip) of BACH1 to bind to PTP1B or IR- β .

Although BTB domain alone is not sufficient to execute the regulation as full-length BACH1, BTB domain of BACH1 mediates the binding of BACH1 with PTP1B or IR- β , as shown in the above response 1.

5. Fig. 1A is difficult to interpret. First, these SNPs can affect nearby genes. Second, it is not shown how significant these SNPs are when viewed genome-wide. Third, authors explain that 10kb regions up and down of BACH1 gene were analyzed. However, BACH1 gene itself spans more than 150 kb DNA. Careful analysis and better, correct figure are needed here.

Response: We thank the reviewer for this important suggestion. We agree that these SNPs can affect nearby genes and there is no solid evidence showing the association of BACH1 gene with these genetic variants linked with the risk of obesity and diabetes. Therefore, the original Fig. 1A and corresponding description were removed from the manuscript.

6. A band can be detected in BACH1 KO samples. Is this a non-specific band?

Response: Thanks a lot for this valuable question. The BACH1 protein was reduced by about 80% in whole-liver lysates from *Bach1*^{LKO} mice (new Fig. S2B). As hepatocytes make up approximately 85% of the total cell population in the liver⁸, the residual BACH1 in whole-liver lysates from *Bach1*^{LKO} mice was likely from non-parenchymal cells, including hepatic stellate cells, Kupffer cells, sinusoidal endothelial cells and pit cells. Therefore, band seen in BACH1 KO liver samples is likely a signal coming from non-hepatic cells present in the liver lysates. We had added this description in the Results of the revised manuscript (line 136-138 on page 7, marked in blue).

Fig. S2. BACH1 is specifically knocked down in the mouse livers.

(B) Representative Western blotting showing the expression of BACH1 in the heart, muscle and liver samples from *Bach1^{fl/fl}* and *Bach1^{LKO}* mice. The columns show the quantification data of BACH1 protein levels of the mouse livers.

7. The regulation of insulin sensitivity in myoblasts and adipocytes are important and results should be shown.

Response: Thanks a lot for this very valuable suggestion. To determine whether BACH1 regulates insulin signaling in myoblasts and adipocytes, BACH1 was overexpressed or knocked down in myoblast C2C12 and pre-adipocyte 3T3-L1 cells. The insulin-stimulated phosphorylation of IR- β , AKT, GSK-3 β , and FOXO1 were examined by Western blot. We observed elevated insulin signaling in both C2C12 cells and 3T3-L1 cells upon insulin stimulation when BACH1 was knocked down by adenovirus vector coding shRNA targeting BACH1 (*shBach1*). The opposite effects were observed in C2C12 and 3T3-L1 cells with BACH1 overexpression induced by the adenovirus vector of BACH1. These results indicate that BACH1 also inhibited insulin signaling in the skeletal muscle cell lines and murine pre-adipocytes.

The new figures are displayed in new supplementary Fig. 7 as follows:

Supplementary Fig 7

Fig. S7. BACH1 regulates insulin signaling in C2C12 and 3T3-L1 cells.

(A) Western blot analysis to detect phosphorylation of key molecules of the insulin pathway in C2C12 cells infected with adenovirus vector coding BACH1 (AdBACH1) (left) or shRNA targeting BACH1 (*shBach1*) (right). **(B)** Western blot analysis to detect phosphorylated key molecules of the insulin pathway in 3T3-L1 cells infected with AdBACH1 (left) or *shBach1*(right). Cells were infected with adenovirus for 48 h and then stimulated with insulin (100 nM) for 10 minutes.

8. Some discussion may be helpful for readers. Does BACH1 protein increase in other disease models?

Response: Thanks a lot for this very valuable suggestion. We have discussed the BACH1 protein increase in other disease models (lines 348-350 on page 16, marked in blue) in the Discussion of the revised manuscript:

“Previous studies have shown that BACH1 expression was significantly increased in hypertrophic human hearts, human carotid and coronary atherosclerotic plaques, and tumors from patients with triple-negative breast cancer and human epithelial ovarian cancer^{9, 10, 11, 12, 13, 14}.”

We are grateful to your effort reviewing our paper and your positive feedback. We have carefully addressed all your concerns. We hope you will be satisfied with our answers and the new data we provided.

References

1. Jais A, *et al.* Heme oxygenase-1 drives metaflammation and insulin resistance in mouse and man. *Cell* **158**, 25-40 (2014).
2. Inoue M, Tazuma S, Kanno K, Hyogo H, Igarashi K, Chayama K. Bach1 gene ablation reduces steatohepatitis in mouse MCD diet model. *J Clin Biochem Nutr* **48**, 161-166 (2011).
3. Tashiro S, *et al.* Repression of PML nuclear body-associated transcription by oxidative stress-activated Bach2. *Mol Cell Biol* **24**, 3473-3484 (2004).
4. Suzuki H, *et al.* Heme regulates gene expression by triggering Crm1-dependent nuclear export of Bach1. *Embo j* **23**, 2544-2553 (2004).
5. Issad T, Boute N, Boubekeur S, Lacasa D. Interaction of PTPB with the insulin receptor precursor during its biosynthesis in the endoplasmic reticulum. *Biochimie* **87**, 111-116 (2005).
6. Boute N, Boubekeur S, Lacasa D, Issad T. Dynamics of the interaction between the insulin

- receptor and protein tyrosine-phosphatase 1B in living cells. *EMBO Rep* **4**, 313-319 (2003).
7. Suzuki H, Tashiro S, Sun J, Doi H, Satomi S, Igarashi K. Cadmium induces nuclear export of Bach1, a transcriptional repressor of heme oxygenase-1 gene. *J Biol Chem* **278**, 49246-49253 (2003).
 8. Mazzocchi A, Devarasetty M, Huntwork R, Soker S, Skardal A. Optimization of collagen type I-hyaluronan hybrid bioink for 3D bioprinted liver microenvironments. *Biofabrication* **11**, 015003 (2018).
 9. Wei X, *et al.* Cardiac-specific BACH1 ablation attenuates pathological cardiac hypertrophy by inhibiting the Ang II type 1 receptor expression and the Ca²⁺/CaMKII pathway. *Cardiovasc Res* **119**, 1842-1855 (2023).
 10. Jia M, *et al.* Deletion of BACH1 Attenuates Atherosclerosis by Reducing Endothelial Inflammation. *Circ Res* **130**, 1038-1055 (2022).
 11. Guo J, *et al.* BACH1 deficiency prevents neointima formation and maintains the differentiated phenotype of vascular smooth muscle cells by regulating chromatin accessibility. *Nucleic Acids Res* **51**, 4284-4301 (2023).
 12. Jiang L, *et al.* Bach1 Represses Wnt/ β -Catenin Signaling and Angiogenesis. *Circ Res* **117**, 364-375 (2015).
 13. Lee J, *et al.* Effective breast cancer combination therapy targeting BACH1 and mitochondrial metabolism. *Nature* **568**, 254-258 (2019).
 14. Han W, *et al.* BTB and CNC homology 1 (Bach1) promotes human ovarian cancer cell metastasis by HMGA2-mediated epithelial-mesenchymal transition. *Cancer Lett* **445**, 45-56 (2019).

Reviewer #2 (Remarks to the Author):

This manuscript revealed the role of BACH1 in insulin sensitivity, and the result demonstrated a novel function for hepatic BACH1 in the regulation of insulin sensitivity and glucose homeostasis, providing a potential target for the treatment of metabolic disorders, the advices were as follow:

Major:

1.The insulin resistance should be detected by Hyperinsulinemic Euglycemic Clamp;

Response: Thank you for this valuable suggestion. To measure the effect of BACH1 deficiency on hepatic insulin sensitivity more accurately, we performed hyperinsulinemic/euglycemic clamp studies in *db/db* diabetic mice (new Fig. 8). We silenced BACH1 by administering AAV-sh*Bach1* to selectively inhibit BACH1 in hepatocytes in *db/db* mice. We injected four-week-old male *db/db* mice with AAV-sh*Bach1* or AAV-sh*Con* to examine whether decreased BACH1 expression could reverse insulin resistance in *db/db* mice. AAV-sh*Bach1* infection caused efficient knockdown of endogenous BACH1 in the liver (Fig. 8I). After the AAV injection, *db/db* mice were fed with chow diet for four weeks. Knockdown of BACH1 in *db/db* mice had no significant effect on body weight and food intake (Fig. S13A-S13B). Consistently with the phenotype of of HFD-fed *Bach1*^{LKO} mice, *db/db* mice injected with AAV-sh*Bach1* exhibited decreased fasting blood glucose, enhanced glucose tolerance, improved hepatic insulin signaling compared to the control group (Fig. 8A-8C, and 8I). To more accurately measure the effect of BACH1 deficiency on insulin sensitivity, we performed hyperinsulinemic/euglycemic clamp studies in *db/db* mice. The glucose infusion rate (GIR), as quantified by the amount of exogenous glucose required to maintain euglycemia, was higher in AAV-sh*Bach1* mice than in the control mice (Fig. 8D). The insulin-stimulated suppression of hepatic glucose production (HGP) was also significantly enhanced in AAV-sh*Bach1* mice when compared to the control mice (Fig 8F). Thus, the hepatic-specific knockdown of BACH1 appeared to reduce blood glucose levels by both increasing insulin sensitivity and reducing hepatic glucose production. Moreover, knockdown of BACH1 in *db/db* mice enhanced the hepatocyte glycogen content and reduced the expression of gluconeogenesis-related genes in the livers of mice (Fig. 8G-8H). Accordingly, we have added the information in Expanded Materials and Methods (line 519-542 on page 24-25, marked in blue) and Results (line 313-333 on page 15-16, marked in blue).

The new figures are displayed in new Fig. 8 as follows:

Fig. 8

Fig. 8. Knockdown of BACH1 ameliorates hyperglycemia and insulin resistance in *db/db* diabetic mice.

(A) Male *db/db* mice were injected with AAV control or AAV-sh*Bach1* for four weeks, the serum fasting blood glucose levels of *db/db* mice were examined (n=8 mice/group). (B) GTTs were measured (n=8 mice/group) (left). The area under the curve (AUC) was used to quantify the GTT results (right). (C) ITTs were measured (n=8 mice/group) (left). The AUC was used to quantify the ITT results (right). (D) The glucose infusion rate (GIR) was calculated for the last 40 minutes of insulin infusion (n=5 mice/group). (E) Basal and clamped rates of hepatic glucose production (HGP) were calculated (n=5

mice/group). **(F)** Suppression of HGP in *db/db* mice receiving either AAV control or AAV-sh*Bach1* was determined (n=5 mice/group). **(G)** A glycogen assay was carried out to determine the glycogen content in the livers of *db/db* mice receiving either AAV control or AAV-sh*Bach1* (n=8 mice/group). **(H)** mRNA expression of *Pck1* and *G6pc* was measured in the livers of *db/db* mice receiving either AAV control or AAV-sh*Bach1* by real-time quantitative PCR (qRT-PCR) (n=6/group). **(I)** Western blot analysis of essential markers of insulin signaling (IR- β , AKT, FOXO1, and GSK-3 β) in the liver tissues from *db/db* mice receiving either AAV control or AAV-sh*Bach1* after injected i.p. with insulin after overnight fasting (n=6/group). **(J)** Working model of the role of hepatic BACH1 in insulin sensitivity regulation. The increased BACH1 expression induced by excessive nutrient intake impaired insulin signaling and aggravated insulin resistance by facilitating the binding of PTP1B and IR- β . Statistical analysis was performed by Student's t test for A-F and H, by Mann-Whitney U test for G.

2. In this manuscript, the in vitro experiment revealed BACH1 interacted with PTP1B and IR- β and its mechanism by immunoprecipitation, GST-pull down and immunofluorescence, but there may be some difference between in vitro and in vivo experiment. We suggested they should investigate BACH1 interacted with PTP1B and IR- β by Small living animal imaging technology in animal models.

Response: Thank you for your important comments. We have carried out experiments to investigate how BACH1 interacted with PTP1B and IR- β by small living animal imaging technology. Unfortunately, the sensitivity of the instrument was not high enough to enable the detection of a clear fluorescence signal. We thus sincerely apologize that we currently are not able to perform this experiment. We have discussed the limitation in the revised manuscript (line 431-432 on page 20, marked in blue).

3. Lipophagy is closely related to insulin resistance, FOXO1 and GSK-3 β . We suggested that some experiments about the effect on lipophagy in BACH1 and its related signal pathways should be supplemented, and also in discussion.

Response: Thanks a lot for this very valuable suggestion. Modulation of liver lipophagy have the potential to ameliorate NAFLD, nonalcoholic steatohepatitis and insulin resistance¹. FOXO1 is an important regulator for Lipophagy². Lipophagy is closely related to insulin resistance, FOXO1, and GSK-3 β ³. To address whether lipophagy was regulated by BACH1, HepG2 cells were transfected by vectors coding for BACH1-HA or control vectors and then treated with oleic acid (OA, 1 mM) for 24 hours to mimic the HFD conditions in vitro. We performed a colocalization study between the autophagosomal marker LC3 and lipid droplets (labelled by BODIPY). Our results showed that BACH1 had little effect on the lipophagy (Supplementary Fig. 8A) and on the protein levels of lipophagy marker PLIN2 and autophagy makers, including LC3, P62, and LAMP1 in OA-treated HepG2 cells (Supplementary Fig. 8B). These results showed that autophagic LDs degradation was not affected by BACH1. Accordingly, we have added the information in Expanded Supplemental Materials (line

156-160 on page 6, marked in blue), Results (line 219-225 on page 11, marked in blue) and Discussion (line 426-429 on page 20, marked in blue).

The new figures are displayed in new Fig. S8 as follows:

Supplementary Fig 8

Fig. S8. Lipophagy was not affected by BACH1

(A) Representative images of lipid droplets (LDs) by BODIPY staining (green) and autophagosomes by LC3 staining (red) in HepG2 cells with or without BACH1 overexpression after treated with 1 mM OA for 24h. Nuclei were stained with DAPI (blue). (B) Western blot analysis of lipophagy-associated protein.

Minor:

The language and grammar style should be polished by native English speaker.

Response: The revised manuscript has been reviewed by a qualified, professional native English editor.

We are grateful to your effort reviewing our paper and your positive feedback. We have carefully addressed all your concerns. We hope you will be satisfied with our answers and the new data we provided.

References

1. Minami Y, *et al.* Liver lipophagy ameliorates nonalcoholic steatohepatitis through extracellular lipid secretion. *Nat Commun* **14**, 4084 (2023).
2. Lettieri Barbato D, Tatulli G, Aquilano K, Ciriolo MR. FoxO1 controls lysosomal acid lipase in adipocytes: implication of lipophagy during nutrient restriction and metformin treatment. *Cell Death Dis* **4**, e861 (2013).
3. He F, *et al.* Mitophagy-mediated adipose inflammation contributes to type 2 diabetes with hepatic insulin resistance. *J Exp Med* **218**, (2021).

Reviewer #3 (Remarks to the Author):

The authors report that BACH1 dampens insulin signaling in the liver, acting beyond its traditional role as a transcription factor. They show that BACH1 expression is increased in several models of obesity in humans and mice. In cells they demonstrate that treatment of hepatocytes with fatty acids induces BACH1 expression, providing a potential link between BACH1 and insulin resistance in an obesogenic environment. This interesting and thorough study sheds light on a novel role of BACH1 in insulin signaling, which may represent a potential mechanism of hepatic insulin resistance.

Major comments:

1) The authors perform a GWAS identifying 94 genetic variants within 5 kb of Bach1 associated with obesity or metabolic disorders. The authors should justify their selection of 5 kb, as this is a very large window and therefore may be less specific to BACH1 effects than say 2 kb. Furthermore, the authors include obesity with other metabolic disorders. Is BACH1 associated with insulin resistance independently of obesity per se? What are the results if obesity is not included as a phenotype? This would give a better idea of the causal effects of BACH1 on metabolic disease, since the authors predict that BACH1 changes in response to obesity.

Response: We thank the reviewer for this important comment. We agree that although there are some genetic variants nearby the BACH1 gene, these SNPs can also affect nearby other genes. Thus, we cannot conclude that BACH1 is associated with the risk of obesity and diabetes. Therefore, we have deleted the original Fig. 1A and corresponding description. In our present study, we found that hepatic expression of BACH1 is significantly upregulated in the livers of obese patients and obese mice, and the treatment of OA and PA upregulated the protein levels of BACH1 in primary hepatocytes. We speculate that hepatic BACH1 might be upregulated by free fatty acids (FFAs) in patients with obesity or mice. In line with this notion, it has been previously reported that a high-fat and high-fructose diet increased BACH1 expression in the livers of mice¹.

A series of studies conducted in mouse models and in humans have demonstrated alterations in adipose tissue biology that link obesity with insulin resistance, and obesity can cause insulin resistance². Most individuals with insulin resistance have obesity, but some individuals can present with significant insulin resistance in muscle and liver in the absence of obesity³. Whether BACH1 is associated with insulin resistance independently of obesity per se is unknown. We do feel that in order to answer this question in full, a thorough and massive sequencing is required to identify the correlation between BACH1 gene and these SNPs in lean individuals with insulin resistance. We will do it in the future work. Thanks a lot for this very valuable suggestion.

2) The authors use HOMA-IR, ITT and GTT as surrogate measures of whole-body insulin sensitivity, and liver glycogen content and gene expression of gluconeogenesis enzymes as liver-specific measures. Unfortunately, direct measures of hepatic insulin sensitivity were not obtained, e.g. rate of hepatic glucose production and/or glycogen synthesis during a hyperinsulinemic euglycemic clamp or in isolated hepatocytes. The authors should justify the use of these indirect measures for liver insulin sensitivity, and tone down the conclusions about a definitive role of BACH1 in hepatic insulin resistance. This should be included in the discussion as a limitation to the study. On a related note, line 248 says BACH1 overexpressing livers have lower glycogen storage “capacity”. However, glycogen storage capacity was not actually measured, only glycogen content. This should be corrected for accuracy.

Response: Thank you for this valuable suggestion. We have replaced “capacity” by “content” in line 248. To measure the effect of BACH1 deficiency on hepatic insulin sensitivity more accurately, we have performed hyperinsulinemic/euglycemic clamp studies in *db/db* diabetic mice (new Fig. 8). We silenced BACH1 by administering AAV-sh*Bach1* to selectively inhibit BACH1 in hepatocytes in *db/db* mice. We injected four-week-old male *db/db* mice with AAV-sh*Bach1* or AAV-shCon to examine whether decreased BACH1 expression could reverse insulin resistance in *db/db* mice. AAV-sh*Bach1* infection caused efficient knockdown of endogenous BACH1 in the liver (Fig. 8I). After the AAV injection, *db/db* mice were fed with chow diet for four weeks. Knockdown of BACH1 in *db/db* mice had no significant effect on body weight and food intake (Fig. S13A-S13B). Consistently with the phenotype of of HFD-fed *Bach1*^{LKO} mice, *db/db* mice injected with AAV-sh*Bach1* exhibited decreased fasting blood glucose, enhanced glucose tolerance, improved hepatic insulin signaling compared to the control group (Fig. 8A-8C, and 8I). To measure the effect of BACH1 deficiency on insulin sensitivity more accurately, we performed hyperinsulinemic/euglycemic clamp studies in *db/db* mice. The glucose infusion rate (GIR), as quantified by the amount of exogenous glucose required to maintain euglycemia, was higher in AAV-sh*Bach1* mice than in the control mice (Fig. 8D). The insulin-stimulated suppression of hepatic glucose production (HGP) was also significantly enhanced in AAV-sh*Bach1* mice when compared to the control mice (Fig 8F). Thus, the hepatic-specific knockdown of BACH1 appeared to reduce blood glucose levels by both increasing insulin sensitivity and reducing hepatic glucose production. Moreover, knockdown of BACH1 in *db/db* mice enhanced the hepatocyte glycogen content and reduced the expression of gluconeogenesis-related genes in the livers of mice (Fig. 8G-8H). Accordingly, we have added the information in Expanded Materials and Methods (line 519-542 on page 24-25, marked in blue) and Results (line 313-333 on page 15-16, marked in blue).

The new figures are displayed in new Fig. 8 as follows:

Fig. 8

Fig. 8. Knockdown of BACH1 ameliorates hyperglycemia and insulin resistance in *db/db* diabetic mice.

(A) Male *db/db* mice were injected with AAV control or AAV-sh*Bach1* for four weeks, the serum fasting blood glucose levels of *db/db* mice were examined (n=8 mice/group). (B) GTTs were measured (n=8 mice/group) (left). The area under the curve (AUC) was used to quantify the GTT results (right). (C) ITTs were measured (n=8 mice/group) (left). The AUC was used to quantify the ITT results (right). (D) The glucose infusion rate (GIR) was calculated for the last 40 minutes of insulin infusion (n=5 mice/group). (E) Basal and clamped rates of hepatic glucose production (HGP) were calculated (n=5 mice/group). (F) Suppression of HGP in *db/db* mice receiving either AAV control or AAV-sh*Bach1* was determined (n=5 mice/group). (G) A glycogen assay was carried

out to determine the glycogen content in the livers of *db/db* mice receiving either AAV control or AAV-sh*Bach1* (n=8 mice/group). **(H)** mRNA expression of *Pck1* and *G6pc* was measured in the livers of *db/db* mice receiving either AAV control or AAV-sh*Bach1* by real-time quantitative PCR (qRT-PCR) (n=6/group). **(I)** Western blot analysis of essential markers of insulin signaling (IR- β , AKT, FOXO1, and GSK-3 β) in the liver tissues from *db/db* mice receiving either AAV control or AAV-sh*Bach1* after injected i.p. with insulin after overnight fasting (n=6/group). **(J)** Working model of the role of hepatic BACH1 in insulin sensitivity regulation. The increased BACH1 expression induced by excessive nutrient intake impaired insulin signaling and aggravated insulin resistance by facilitating the binding of PTP1B and IR- β . Statistical analysis was performed by Student's t test for A-F and H, by Mann-Whitney U test for G.

3) The authors show that BACH1 blunts insulin signaling in hepatocytes through interaction of its BTB domain with insulin receptor β and the phosphatase PTP1B. However, these experiments were performed at only one (maximal) dose of insulin (100 nM). Because submaximal insulin doses were not used as well, from a pharmacological perspective it is not definitive whether insulin sensitivity or max response is shifted. This should be specified in the manuscript.

Response: Thank you for your valuable suggestions. As suggested, we have performed additional experiments to treat hepatocytes from low to high insulin concentrations (25~200 nM). The results showed that HepG2 cells infected with adenoviruses AdBACH1 had weakened and delayed phosphorylation of IR- β , AKT, and GSK-3 β at multiple insulin concentrations (New Fig. 4C). Therefore, these data indicate that BACH1 reduced the insulin sensitivity. We have added it to the Results section (page 10, line 211-214) in the revised manuscript.

The new figures are displayed in Fig. 4C as follows:

Fig. 4. BACH1 inhibits insulin signaling in hepatocytes.

(C) Left: Phosphorylation of key molecules of the insulin pathway was determined in the HepG2 cells infected with adenoviruses encoding AdBACH1 or AdGFP and then stimulated with different concentrations of insulin for 10 minutes. Right: The phosphorylated protein levels normalized to total protein (n=3/group).

4) The authors show that BACH1 modifies insulin signaling, and that modification of BACH1 expression affects surrogate measures of insulin sensitivity, at least partly dependent on PTP1B as shown by PTP1B knockdown. While these experiments are elegant and informative, I do not believe that they definitively link changes in hepatocyte insulin signaling with changes in insulin sensitivity in the AAV8-TBG mouse models. For example, what are coincidental changes in liver protein expression as a result of the actions of BACH1 as a transcription factor? Does deletion of the BTB domain affects its activity as a transcription factor? This should be discussed.

Response: We thank you for this important comment. We have performed hyperinsulinemic/euglycemic clamp studies to demonstrate that BACH1 deficiency increased hepatic insulin sensitivity in *db/db* mice (new Fig. 8), and further confirmed that modification of BACH1 expression affected insulin sensitivity *in vivo*. Because HO-1 is a result of the actions of BACH1 as a transcription factor, we have performed additional experiments to determine HO-1 expression in BACH1 KO hepatocytes. We compared the expression of insulin signaling and HO-1 by expressing WT BACH1, BACH1 lacking BTB, and BACH1 lacking Bzip in BACH1 KO hepatocytes from *Bach1*^{LKO} mice. It was reported that the N-terminal region of BACH1 contains a BTB domain, which functions as a protein interaction motif. The C-terminal BZip domain of BACH1 binds to DNA and mediates the heterodimerization of BACH1 with small Maf proteins⁴. We found that BACH1 deficiency led to elevated phosphorylation levels of IR- β , AKT, GSK-3 β , and FOXO1 in primary hepatocytes isolated from *Bach1*^{LKO} mice upon insulin stimulation, which were significantly inhibited by full BACH1 (BACH1-HA) and truncated vector lacking the Bzip domain (BACH1 ^{Δ Bzip}-HA), but not by vector lacking the N-terminal BTB domain of BACH1 (BACH1 ^{Δ BTB}-HA) (new Fig. 5A). These results suggested that the BTB domain, but not the Bzip domain of BACH1, is essential for repressing the insulin signaling. We also found that HO-1 expression in BACH1 KO cells was significantly inhibited by full BACH1, but not by BACH1 ^{Δ Bzip} or BACH1 ^{Δ BTB}, which suggests that both the BTB domain and the Bzip domain of BACH1 are important for the regulation of HO-1. These findings indicate that deletion of the BTB domain of BACH1 also affects its activity as a transcription factor. It was reported that hepatocyte conditional HO-1 deletion in mice evoked resistance to diet-induced insulin resistance and reduces diet-induced fatty liver disease⁵. *Bach1* ablation exerts hepatoprotective effect against steatohepatitis presumably via HO-1 induction⁶. Thus, the transcriptional regulatory effect of BACH1 may also be involved in regulating insulin signaling, and we cannot rule out this possibility. The transcription regulation may be involved in the function of BACH1 in the regulation of insulin signaling. We have made corresponding modifications in the entire manuscript and added the aforementioned descriptions to the Discussion section (lines 388-394 on page 18, marked in blue) in the revised manuscript.

The new figure is displayed in new Fig. 5A as follows:

Fig.5

Fig. 5. BACH1 interacts with PTP1B and IR- β by the BTB domain.

(A) Left: Western blot analysis of the phosphorylation and total protein levels of IR- β , AKT, and GSK-3 β in primary hepatocytes isolated from *Bach1*^{LKO} mice. Hepatocytes were transfected with vectors coding for HA-tagged versions of the full *Bach1* sequence (BACH1-HA) or mutant sequences lacking the Bzip domain (BACH1 Δ Bzip-HA) or BTB domain (BACH1 Δ BTB-HA) with or without insulin (100 nM) stimulation for 10 min. Right: Phosphorylated protein levels were normalized to total protein.

5) The authors overexpress BACH1 in liver by tail vein injection of AAV8-TBG-Cre into BACH1 fl/fl mice. To delete BACH1, AAV8-Bach1-shRNA is injected. AAV8-TBG is known to have good hepatocyte specificity. Indeed the authors show total protein levels of BACH1 are unaffected in several other tissues, only in liver. AAV8-TBG-GFP is used as a negative control, which is appropriate since AAV8 is known to elicit an immune response in hepatocytes. The authors should perform IHC of BACH1 in liver sections to demonstrate Cre efficiency (% of cells and location, e.g. is the periportal region more affected and periphery less affected).

Response: Thank you for your valuable suggestions. As suggested, we have performed IHC of BACH1 in liver sections to demonstrate Cre efficiency (Fig. S2C). The results showed BACH1 was mainly expressed in liver sinusoidal endothelial cells and hepatocytes in livers from WT mice, while about 78% of hepatocytes showed no BACH1 staining, especially in the portal area and surrounding zones in livers from hepatocyte-specific *Bach1* knockout mice. Hepatic overexpression of *Bach1* in mice was validated by immunohistochemistry, and BACH1 was specifically overexpressed in hepatocytes (Fig. S4C).

The new figures are displayed in new Fig. S2C and S4C as follows:

Fig. S2. BACH1 is specifically knocked down in the mouse livers.

(C) The representative images of immunohistochemistry assay of BACH1 in *Bach1*^{fl/fl} and *Bach1*^{LKO} mouse liver samples. Right: Quantitative data of immunohistochemistry assay of BACH1 in the liver tissues (n=3).

Fig. S4. BACH1 is specifically overexpressed in the mouse livers.

(C) The representative images of immunohistochemistry assay of BACH1 in NTG and *Bach1*^{LTG} mouse liver samples.

6) For the glucose tolerance test, the authors perform an I.P. GTT rather than oral gavage. They should list this as a limitation, since I.P. GTT causes a poor insulin secretory response compared to oral gavage, and therefore may be less relevant as a measure of insulin sensitivity. Furthermore, the tolerance tests are dosed by body weight rather than lean mass, which may confound interpretation as obese animals will have a higher delivery of glucose or insulin than lean animals.

Response: We thank you for this important comment. We have mentioned this as a study limitation. Previous studies found the tolerance tests dosed by body weight, lean mass and fixed dose detected a similar decrease in insulin sensitivity in DIO mice⁷. However, the different glucose dosing regimens gave results regarding glucose elimination and the acute insulin response. The fixed-dose regimen was the only that revealed impairment of glucose elimination, whereas dosing according to total BW was the only regimen which showed significant increases in acute insulin response in DIO mice. Furthermore, blood glucose AUC was higher increased in obese mice when the glucose dose was calculated according to total body weight⁷, which may confound interpretation as obese animals will have a higher delivery of glucose or insulin than lean animals. As suggested, we have added this in the Discussion and Results (line

432-433, page 20; line 324-325, page 15; line 332-333, page 16) as following: “Thirdly, intraperitoneal GTT may cause a poor insulin secretory response compared to oral gavage”. “To measure the effect of BACH1 deficiency on insulin sensitivity more accurately, we performed hyperinsulinemic/euglycemic clamp studies in db/db mice”, “the hepatic-specific knockdown of BACH1 appeared to reduce blood glucose levels by both increasing insulin sensitivity and reducing hepatic glucose production”.

7) The mouse experiments were performed in males only. The authors should justify their choice of males and exclusion of females. This should be listed in the discussion as a study limitation.

Response: Thanks for the reviewer's valuable suggestion. We apologize for the absence of data on female mice. This is indeed a limiting factor in our study, and we have clearly stated this as a study limitation in our discussion (line 430-431 on page 20, marked in blue) in the revised manuscript.

At partial justification, male subjects are more likely to develop abnormal liver fat accumulation, non-alcoholic fatty liver disease, liver fibrosis, and liver tumors compared with female subjects, before the menopause^{8, 9, 10}. Therefore, we started with testing male mice in our study.

Minor comments:

1) Fig 6B magnification too small, inset magnification too small

Response: Thank you for your comments. In the revised manuscript, we have repeated the immunofluorescence assay and modified Fig 6B.

The modified fig 6B are displayed as follows:

Fig. 6. BACH1 enhances the interaction between PTP1B and IR-β in response to insulin.

(B) Representative immunostaining of IR-β-cherry (red), PTP1B-D181A-BFP (purple), and GFP (green) in HepG2 cells with or without BACH1-GFP overexpression after treated with 100 nm insulin for 10 min. Nuclei were stained with DAPI (blue) (upper). Colocalization rate were calculated and showed (lower). Scale bars=20 μm.

2) Fig 7G BACH1 blot overexposed, cropped too tightly

Response: We thank for this very valuable suggestion. As requested, we have replaced the BACH1 blot in the revised Fig 7G as following.

Fig. 7. BACH1 aggravates insulin resistance through a PTP1B-dependent manner.

(G) Left: Western blot analysis of essential markers of the insulin pathway (IR-β, AKT, FOXO1, and GSK-3β) in the livers from the above 4 groups of mice after insulin administration. Right: Phosphorylated protein levels were normalized to total protein (n=6 mice/group).

3) Fig 8 working model too much interpretation in figure legend—“These findings suggest that BACH1 could be an attractive therapeutic target for the treatment of diabetes, potentially facilitating improved insulin sensitivity and glucose tolerance.”

Response: We thank you a lot for this comment. We have deleted the sentence “These findings suggest that BACH1 could be an attractive therapeutic target for the treatment of diabetes, potentially facilitating improved insulin sensitivity and glucose tolerance.”, and streamlined the interpretation in the figure legend of Fig 8J as following: “The increased BACH1 expression induced by excessive nutrient intake impaired

insulin signaling and aggravated insulin resistance by facilitating the binding of PTP1B and IR- β ”.

4) Line 394—*Bach1* should be capitalized

Response: Thank you. We have capitalized “*Bach1*” as “BACH1” in the original 394 lines, and 400 lines of revised manuscript.

5) Line 429—should read “of BACH1”

Response: We have corrected “*Bach1*” as “of BACH1” in the original line 429, and line 420 of revised manuscript.

6) Line 849—should be “In brief” or “Briefly”

Response: We are sorry about the grammatical errors and we have made corrections. “In briefly” has been modified to “In brief” in the original line 849, and now line 119 of Supplemental Materials.

We are grateful to your effort reviewing our paper and your positive feedback. We have carefully addressed all your concerns. We hope you will be satisfied with our answers and the new data we provided.

References

1. Ka SO, Bang IH, Bae EJ, Park BH. Hepatocyte-specific sirtuin 6 deletion predisposes to nonalcoholic steatohepatitis by up-regulation of Bach1, an Nrf2 repressor. *Faseb j* **31**, 3999-4010 (2017).
2. Klein S, Gastaldelli A, Yki-Järvinen H, Scherer PE. Why does obesity cause diabetes? *Cell Metab* **34**, 11-20 (2022).
3. James DE, Stöckli J, Birnbaum MJ. The aetiology and molecular landscape of insulin resistance. *Nat Rev Mol Cell Biol* **22**, 751-771 (2021).
4. Oyake T, *et al.* Bach proteins belong to a novel family of BTB-basic leucine zipper transcription factors that interact with MafK and regulate transcription through the NF-E2 site. *Mol Cell Biol* **16**, 6083-6095 (1996).
5. Jais A, *et al.* Heme oxygenase-1 drives metaflammation and insulin resistance in mouse and man. *Cell* **158**, 25-40 (2014).
6. Inoue M, Tazuma S, Kanno K, Hyogo H, Igarashi K, Chayama K. Bach1 gene ablation reduces steatohepatitis in mouse MCD diet model. *J Clin Biochem Nutr* **48**, 161-166

(2011).

7. Jørgensen MS, Tornqvist KS, Hvid H. Calculation of Glucose Dose for Intraperitoneal Glucose Tolerance Tests in Lean and Obese Mice. *J Am Assoc Lab Anim Sci* **56**, 95-97 (2017).
8. Mosca L, Barrett-Connor E, Wenger NK. Sex/gender differences in cardiovascular disease prevention: what a difference a decade makes. *Circulation* **124**, 2145-2154 (2011).
9. Tramunt B, *et al.* Sex differences in metabolic regulation and diabetes susceptibility. *Diabetologia* **63**, 453-461 (2020).
10. Rani A, *et al.* Quinolinylnyl β -enaminone derivatives exhibit leishmanicidal activity against *Leishmania donovani* by impairing the mitochondrial electron transport chain complex and inducing ROS-mediated programmed cell death. *J Antimicrob Chemother*, (2022).

Reviewer #4 (Remarks to the Author):

This is a very interesting paper showing original results of the role of BACH1 protein in insulin signalling and glucose homeostasis in the liver of mice. The authors used different strategies (knockout, gain-of-function, cell experiments) to demonstrate that BACH1, through its interaction with PTP1B and the insulin receptor beta, can inhibit insulin stimulation in hepatocytes and reduce glucose tolerance, especially under a high fat diet condition. Although their results are convincing about the important role that may have BACH1 in glucose metabolism in mice, I have concern about supporting evidence in humans.

1. First of all, there is no significant SNPs at the traditional GWAS p-value threshold in the BACH1 gene region. The threshold that should be used is $P < 5 \times 10^{-8}$. All SNPs in Fig.1 are below that threshold. Are there other relevant results in the GWAS Catalog for that gene?

Response: Thank you for your valuable suggestions. Since there is no significant SNPs at the traditional GWAS p-value threshold ($P < 5 \times 10^{-8}$) in the BACH1 gene region, and there are no other relevant results in the GWAS Catalog for BACH1, we have deleted original Fig. 1A and corresponding description.

2. Again regarding human results, I am not sure we can say that the expression of BACH1 in the liver of metabolically healthy and unhealthy are significantly different since an outlier seems to bias the results.

Response: Thanks a lot for these valuable comments. We have performed statistical analysis and compared the BACH1 expression with or without the outlier. The results remain significant with or without the outlier included in the analysis between metabolically healthy and unhealthy groups. Nevertheless, we have analyzed another published bulk RNA-seq data of liver samples from lean or obese individuals (GEO accession no. GSE192742)¹. Notably, the expression of BACH1 was higher in the hepatocytes of obese subjects than in lean subjects (Fig. 1A). To further validate our results, we have performed qRT-PCR to detect BACH1 mRNA expression in the livers of HFD mice and NAFLD patients, as well as Western blot analysis of hepatic BACH1 in patients. As shown in new Fig. 1B-1C and 1G, BACH1 protein and mRNA expression are significantly upregulated in the livers of NAFLD patients as well as HFD mice.

The new figures are displayed in new Fig. 1A-1C and 1G as follows:

Fig. 1

Fig. 1. BACH1 is elevated in the livers of patients with NAFLD and obese mice.

(A) *BACH1* mRNA expression in the livers of lean or obese individuals in the public RNA-sequencing database (GEO accession no. GSE192742). (B-C) *BACH1* mRNA expression (B) and protein expression (C) in the liver tissues from healthy subjects (Normal) and NAFLD patients (n=6). (D) The representative images of H&E, PAS staining, and immunohistochemistry assay of BACH1 from liver tissues of healthy subjects and NAFLD patients. Scale bars=100 μ m. (E) Quantitative data of immunohistochemistry assay of BACH1 in the liver tissues (n=6). (F) *Bach1* mRNA expression in liver cells from mice following a high-sugar diet (HSD) compared with

chow diet (CD) (GEO accession no. GSE182365). **(G-H)** *Bach1* mRNA **(G)** (n=8) and protein expression **(H)** (n=5) in the liver tissues from male CD- and HFD-fed mice. **(I-J)** BACH1 protein expression in male *ob/ob* mice **(I)** (n=4) and *db/db* mice **(J)** (n=3). **(K)** Primary hepatocytes were treated with oleic acid (OA, 1 mM) for 12 hours and then subjected to immunoblot analyses to determine the BACH1 protein expression (n=3). Statistical analysis was performed by Mann-Whitney U test for A, E, G, and by Student's t test for B, F.

3. Since the results shown in human are not convincing, I would try to add more human results published in the literature regarding the potential role of BACH1 in metabolic disorders, if there is more than what was said in the paper (introduction, discussion).

Response: Thanks a lot for the valuable suggestion. We have added human results published in the literature regarding the potential role of BACH1 in metabolic disorders in the Discussion (reproduced below, line 356-358, page17).

“Significant upregulation of BACH1 was also observed in human islets exposed to palmitate, as well as human glomerular endothelial cells and the aorta of diabetic rats upon high glucose^{2, 3, 4}”

4. Minor comment: Maybe use ‘BACH1 Regulates Insulin Signalling’ rather than ‘Resistance’ in the running title.

Response: We thank you for this very valuable suggestion. As suggested, we have replaced running title with ‘BACH1 Regulates Insulin Signaling’.

We are grateful to your effort reviewing our paper and your positive feedback. We have carefully addressed all your concerns. We hope you will be satisfied with our answers and the new data we provided.

Reference

1. Guilliams M, *et al.* Spatial proteogenomics reveals distinct and evolutionarily conserved hepatic macrophage niches. *Cell* **185**, 379-396.e338 (2022).
2. Lytrivi M, *et al.* Combined transcriptome and proteome profiling of the pancreatic β -cell response to palmitate unveils key pathways of β -cell lipotoxicity. *BMC Genomics* **21**, 590 (2020).
3. Li X, *et al.* The SETD8/ELK1/bach1 complex regulates hyperglycaemia-mediated EndMT in diabetic nephropathy. *J Transl Med* **20**, 147 (2022).
4. Meng Z, *et al.* Bach1 modulates AKT3 transcription to participate in hyperglycaemia-mediated EndMT in vascular endothelial cells. *Clin Exp Pharmacol Physiol* **50**, 443-452 (2023).

REVIEWERS' COMMENTS

Reviewer #1 (Remarks to the Author):

This manuscript appears interesting and important to the fields of metabolism, with a completely new mechanism of insulin signal transduction regulation by transcription factor BACH1. Very surprisingly, the regulation did not involve transcription regulation. Issues raised by this reviewer have been effectively addressed by adding new experiments. Congratulations on the beautiful work done.

Reviewer #2 (Remarks to the Author):

This manuscript should be accepted in this style

Reviewer #3 (Remarks to the Author):

I thank the authors for their thoughtful and thorough responses and additional experiments. I have no further comments.

Response to Reviewers

Manuscript No. NCOMMS-23-02730A R2

**“BACH1 Controls Hepatic Insulin Signaling and Glucose Homeostasis”
by Dr. Jiayu Jin et al.**

Reviewer #1

Remarks to the Author:

This manuscript appears interesting and important to the fields of metabolism, with a completely new mechanism of insulin signal transduction regulation by transcription factor BACH1. Very surprisingly, the regulation did not involve transcription regulation. Issues raised by this reviewer have been effectively addressed by adding new experiments. Congratulations on the beautiful work done.

Response: We would like to thank you again for taking the time to review our manuscript and your positive feedback.

Reviewer #2

Remarks to the Author:

This manuscript should be accepted in this style

Response: We are grateful to your effort reviewing our paper and your positive feedback.

Reviewer #3

Remarks to the Author:

I thank the authors for their thoughtful and thorough responses and additional experiments. I have no further comments.

Response: We would like to thank you again for taking the time to review our manuscript and your positive feedback.